# ADAMAS: HADAMARD SPARSE ATTENTION
# FOR EFFICIENT LONG-CONTEXT INFERENCE

## ABSTRACT

Large language models (LLMs) now support context windows of hundreds of thousands to millions of tokens, enabling applications such as long-document summarization, large-scale code synthesis, multi-document question answering and persistent multi-turn dialogue. However, such extended contexts exacerbate the quadratic cost of self-attention, leading to severe latency in autoregressive decoding. Existing sparse attention methods alleviate these costs but rely on heuristic patterns that struggle to recall critical key-value (KV) pairs for each query, resulting in accuracy degradation. We introduce **Adamas**, a lightweight yet highly accurate sparse attention mechanism designed for long-context inference. Adamas applies the Hadamard transform, bucketization and 2-bit compression to produce compact representations, and leverages Manhattan-distance estimation for efficient top-$k$ selections. Experiments show that Adamas matches the accuracy of full attention with only a 64-token budget, achieves near-lossless performance at 128, and supports up to $8\times$ higher sparsity than prior state-of-the-art (SOTA) methods while delivering up to $4.4\times$ self-attention and $1.5\times$ end-to-end speedups on 32K-length sequences. Remarkably, Adamas attains comparable or even lower perplexity than full attention, underscoring its effectiveness in maintaining accuracy under aggressive sparsity. Code is publicly available at https://anonymous.4open.science/r/Adamas-36EA.

## 1 INTRODUCTION

The rapid progress of LLMs has dramatically extended their affordable context windows. Contemporary systems such as Anthropic's Claude Sonnet 4 support up to 1M tokens (Anthropic, 2025), while OpenAI's GPT-5 effectively handles contexts of 128K–256K tokens (OpenAI, 2025). These expanded capacities enable advanced applications, including long-document summarization (Chang et al., 2024), large-scale code synthesis (Nijkamp et al., 2023; Dainese et al., 2024), multi-document question answering (Wu et al., 2025a; Wang et al., 2024b), and persistent multi-turn dialogue (Wu et al., 2025b). By processing extensive information without manual segmentation, LLMs are increasingly able to tackle tasks that demand global coherence and long-term memory. However, these impressive expansions come with a cost. The quadratic complexity of self-attention (Zaheer et al., 2020; Vaswani et al., 2017), together with the scaling of the KV cache, leads to significant per-token latency and memory overhead.

Sparse attention offers a promising solution by restricting each query to attend only a carefully selected subset of tokens (Yuan et al., 2025). This reduces the number of KV pairs involved in attention computation, lowering both computational complexity and memory access costs while largely preserving modeling capacity. Nevertheless, existing approaches often fall into two categories with notable limitations. Static sparsity patterns (e.g., StreamingLLM (Xiao et al., 2023)) are often hand-designed from empirical attention heatmaps, yielding fixed patterns, such as fixed local windows or vertical stripes. However, these patterns fail to capture the dynamic nature of query–key interactions, leading to low recall and degraded accuracy (See Table 2). In contrast, dynamic methods like Quest (Tang et al., 2024) adapt token selection during inference, but their page-level granularity remains overly coarse. This coarse selection introduces token redundancy and limits the achievable sparsity ratio, since higher sparsity levels lead to accuracy degradation.

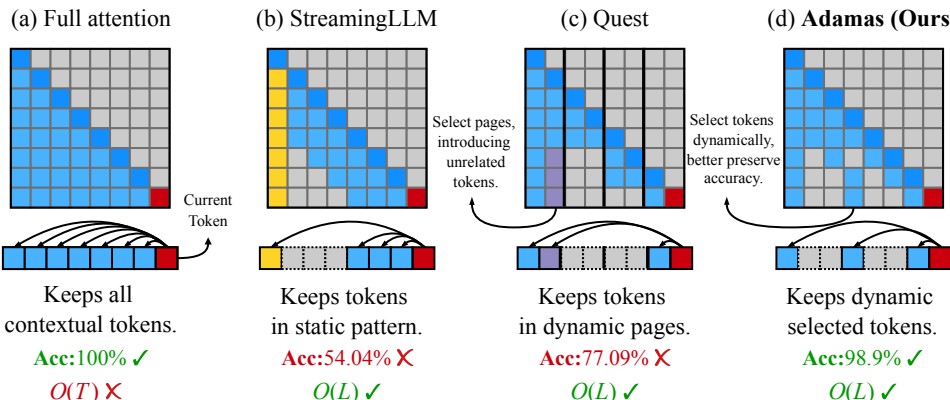

(a) Full attention
(b) StreamingLLM
(c) Quest
(d) **Adamas (Ours)**

Keeps all contextual tokens.
**Acc:**100% ✓
$O(T)$ ✗

Keeps tokens in static pattern.
**Acc:**54.04% ✗
$O(L)$ ✓

Keeps tokens in dynamic pages.
**Acc:**77.09% ✗
$O(L)$ ✓

Keeps dynamic selected tokens.
**Acc:**98.9% ✓
$O(L)$ ✓

Figure 1: Illustration of Adamas compared with existing methods. While StreamingLLM employs a fixed sparse pattern and Quest inherently selects pages, Adamas dynamically selects KV pairs at the token level, thereby achieving better preservation of model accuracy and inference efficiency.[1]

In this paper, we propose Adamas, a lightweight yet highly accurate token-level sparse attention mechanism for long-context inference. Adamas achieves high sparsity while maintaining the ~~attention quality~~ model accuracy, as illustrated in Figure 1. ~~Inspired by QuaRot (Ashkboos et al., 2024), which uses the Hadamard transform to suppress outlier features, we extend this idea by combining the Hadamard transform with bucketization to efficiently approximate query-key similarity at token-level.~~ Inspired by the general idea of using orthogonal transforms to stabilize feature distributions, we incorporate the classical Hadamard transform into Adamas to uniformly redistribute information across dimensions, thereby improving the robustness of our bucketization-based similarity estimation at the token level. In Adamas, queries and keys are first transformed by the Hadamard transform and then compressed into 2-bit codes via bucketization, which are stored in the KV cache with negligible overhead. During decoding, candidate keys are rapidly pre-selected using a lightweight Manhattan-distance estimator on the compressed codes, followed by top-$k$ filtering and sparse attention over the reduced candidate set. The proposed Adamas delivers substantial efficiency gains while preserving attention quality comparable to dense methods. We evaluate both the accuracy and efficiency of Adamas. Due to the dynamical selection of the most relevant KV pairs for each query, Adamas achieves up to $8\times$ higher sparsity than previous SOTA methods while preserving comparable accuracy with full attention under a constrained budget of 128 tokens. Comprehensive evaluations demonstrate that Adamas delivers up to $4.4\times$ self-attention and $1.5\times$ end-to-end speedups on 32K-length sequences, accompanied by perplexity that is even lower than that of full attention, outperforming prior SOTA methods by more than $2\times$. The main contributions of this work are as follows:

- We propose Adamas, a novel sparse-attention mechanism that integrates Hadamard transform, bucketization, 2-bit compression and Manhattan-distance estimator, achieving $8\times$ higher sparsity than prior SOTA methods while preserving comparable accuracy with full attention under constrained budget of 128 tokens.
- We develop high-performance GPU kernels for Adamas, featuring fused bucketization and 2-bit compression together with a lightweight Manhattan-distance estimator, enabling up to $4.4\times$ self-attention and $1.5\times$ end-to-end speedups over full attention in long-context decoding.
- Through extensive ablation studies, we validate the effectiveness of each component in Adamas, demonstrating that every step, including Hadamard transform, bucketization, compression, and estimation, contributes to its overall performance and efficiency.

---

[1]Accuracy (Acc) results are derived from the normalized performance of our evaluations on LongBench. Detailed results can be found in Figure 3 and Table 1.

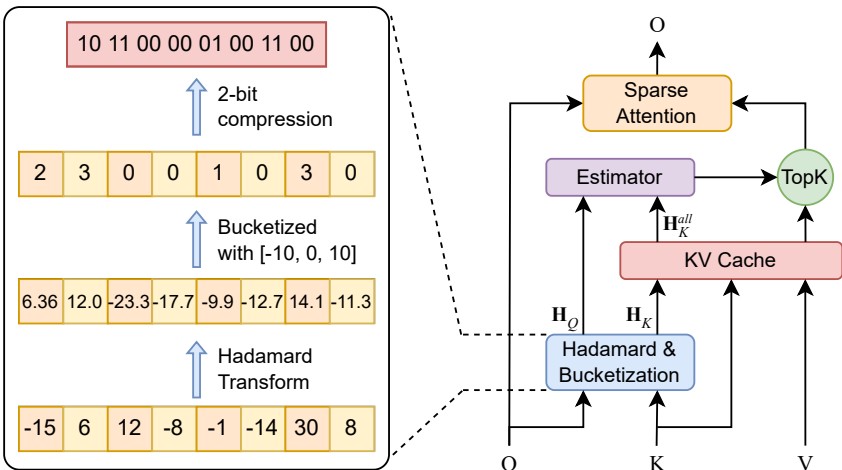

Figure 2: Overview of Adamas. Queries $Q$ and keys $K$ are processed through Hadamard transform, bucketization, and 2-bit compression. The transformed keys $\mathbf{H}_K$ are then compared against the transformed query $\mathbf{H}_Q$ in Manhattan-distance estimator, based on which the top-$k$ KV pairs are selected. Finally, Adamas performs sparse attention using $Q$ and the selected KV pairs.

## 2 PRELIMINARIES

In this section, we briefly review sparse attention and the Hadamard transform to provide background and facilitate a clearer understanding of our proposed method.

**Sparse attention** is a variant of the self-attention mechanism. The core process is defined as

$$O = \text{Softmax}\left(\frac{QK^\top}{\sqrt{d}}\right) V, \tag{1}$$

where $Q$, $K$, $V$, and $O$ denote the query, key, value, and attention output, respectively, and $d$ is the head dimension. Since $QK^\top$ computes the query–key similarity, achieving high recall requires selecting the most relevant keys for each query. However, evaluating $QK^\top$ against all keys incurs quadratic complexity of sequence length, making it impractical for long contexts. Sparse attention addresses this challenge by efficiently approximating $QK^\top$ with only a carefully chosen subset of keys, thereby reducing both computational and memory costs while preserving the model's accuracy.

**Hadamard transform**, also known as the Walsh–Hadamard transform, is an orthogonal linear transform that projects the vector to a new basis defined by Walsh functions. The Hadamard transform corresponds to multiplication by a Hadamard matrix $\mathbf{H_d}$, which is a square matrix of order $d = 2^n$ with entries restricted to $\{+1, -1\}$. The smallest Hadamard matrix is defined as:

$$\mathbf{H}_2 = \frac{1}{\sqrt{2}} \begin{bmatrix} 1 & 1 \\ 1 & -1 \end{bmatrix} \tag{2}$$

Higher-order matrices are constructed recursively via the Kronecker product $\mathbf{H}_{2^n} = \mathbf{H}_2 \otimes \mathbf{H}_{2^{n-1}}$. This recursive structure enables an efficient Hadamard transform algorithm that computes the matrix–vector product $\mathbf{H}x$ in $\mathcal{O}(d \log_2 d)$ rather than the naive $\mathcal{O}(d^2)$. For dimensions $d$ that are not exact powers of two, the existence of a Hadamard matrix is not guaranteed. In such cases, one may exploit factorizations $d = 2^n m$, where $m$ is the order of a known Hadamard matrix, and apply the Kronecker construction $\mathbf{H}_d = \mathbf{H}_{2^n} \otimes \mathbf{H}_m$, yielding a transform with complexity $\mathcal{O}(d(m + n))$.

## 3 METHOD

In this section, we present Adamas, an efficient variant of the Transformer sparse attention mechanism that could substantially reduce computation overhead while preserving model accuracy. We show the workflow of Adamas in Figure 2.

---

**Algorithm 1** Workflow of Adamas

---

**Input:** Query $\mathbf{Q}$, Key $\mathbf{K}$, and Value $\mathbf{V} \in \mathbb{R}^d$, Hadamard matrix $\mathbf{H} \in \mathbb{R}^{d \times d}$
**Output:** Attention output matrix $\mathbf{O}$
  1: $\mathbf{H}_Q \leftarrow \mathbf{QH};$   $\mathbf{H}_K \leftarrow \mathbf{KH}$                                ▷ Apply Hadamard transform
  2: $\widehat{\mathbf{H}}_Q, \widehat{\mathbf{H}}_K \leftarrow \text{Bucketize}(\mathbf{H}_Q, \mathbf{H}_K)$              ▷ Quantize into $\{0, 1, 2, 3\}^d$
  3: $\widehat{\mathbf{H}}_Q, \widehat{\mathbf{H}}_K \leftarrow \text{Compress}(\widehat{\mathbf{H}}_Q, \widehat{\mathbf{H}}_K)$                  ▷ Pack into 2-bit codes
  4: $\widehat{\mathbf{H}}_K^{all}, \mathbf{K}^{all}, \mathbf{V}^{all} \leftarrow \text{KVCache.update}(\widehat{\mathbf{H}}_K, \mathbf{K}, \mathbf{V})$
  5: $D \leftarrow \text{ManhattanDistance}(\widehat{\mathbf{H}}_Q, \widehat{\mathbf{H}}_K^{all})$       ▷ Estimate query–key similarity
  6: $I \leftarrow \text{Top-}k(D)$                                    ▷ Select top-$k$ candidate indices
  7: $\mathbf{K}^s, \mathbf{V}^s \leftarrow \mathbf{K}^{all}[I], \mathbf{V}^{all}[I]$
  8: $\mathbf{O} \leftarrow \text{SparseAttenion}(\mathbf{Q}, \mathbf{K}^s, \mathbf{V}^s)$
  9: **return O**

---

### 3.1 MOTIVATIONS

The design of Adamas is motivated by the dual goals of theoretical equivalence and practical efficiency: leveraging the mathematical equivalence of Hadamard-transformed similarities with the original attention formulation, while exploiting the smoothing property of the transform to enable effective low-bit quantization and lightweight similarity estimation.

**Theoretical perspective.** The Hadamard transform is an orthogonal transformation, which ensures that the similarity computation in the Hadamard domain is mathematically equivalent to that in the original space. Specifically, for queries $Q$ and keys $K$, we have

$$(Q\mathbf{H})(K\mathbf{H})^{\top} = Q(\mathbf{H}\mathbf{H}^{\top})K^{\top} = QK^{\top} \tag{3}$$

where $\mathbf{H}$ denotes a Hadamard matrix with $\mathbf{H}\mathbf{H}^{\top} = \mathbf{I}$. This equivalence shows that applying the Hadamard transform to both $Q$ and $K$ does not incur any information loss.

**Practical perspective.** In practice, directly quantizing query and key vectors can incur severe information loss due to the presence of large-magnitude outlier values, which dominate the distribution. The Hadamard transform mitigates this issue by redistributing variance more evenly across dimensions and suppressing extreme outliers, thereby producing smoother value distributions (Elhage et al., 2023; Ashkboos et al., 2024). As a result, bucketization after the Hadamard transform can approximate the original similarity structure with minimal degradation, enabling compact 2-bit representations that significantly reduce memory overhead while retaining sufficient accuracy. Moreover, since the Hadamard matrix contains only $\pm 1$ entries, its application can be implemented via fast Hadamard transform (Dao, 2024; IBM & Meta, 2024) rather than costly dense matrix multiplications, further reducing computational overhead.

### 3.2 ADAMAS

As shown in Algorithm 1, Adamas modifies the standard Transformer attention pipeline with three innovations: 1) Hadamard transform applied to queries and keys (yielding what we call Hadamard vectors), 2) bucketization and 2-bit compression for Hadamard vectors, and 3) a Manhattan-distance estimator for candidate token selection. These components work jointly to enable faster attention computation with modest memory cost and near lossless accuracy degradation.

**Hadamard transform applied to queries and keys.** Formally, given queries and keys $Q, K \in \mathbb{R}^d$, we compute their Hadamard-transformed representations as

$$H_Q = Q\mathbf{H}, \qquad H_K = K\mathbf{H}, \tag{4}$$

where $\mathbf{H} \in \mathbb{R}^{d \times d}$ denotes a Hadamard matrix.

**Bucketization and 2-bit compression.** For efficient computation and storage, each element in $H_Q, H_K \in \mathbb{R}^d$ is bucketized into one of four levels using predefined thresholds $\{B_1, B_2, B_3\}$, mapped to the discrete set $\{0, 1, 2, 3\}$, which can be encoded into 2-bit integer. The bucketization operator $B(\cdot)$ is defined as

$$B(x) = \sum_i \mathbb{I}(x > B_i), \tag{5}$$

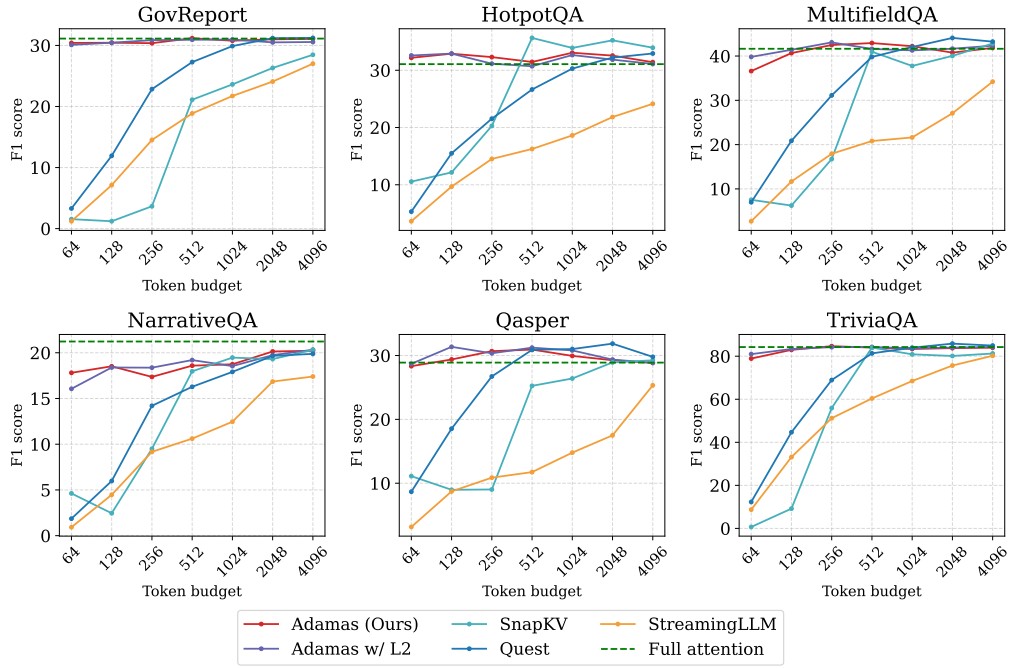

Figure 3: Evaluation results on LongBench. Adamas exhibits the smallest performance drop compared to full attention while maintaining high sparsity.

where $\mathbb{I}(\cdot)$ denotes the indicator function, which evaluates to 1 if the condition inside holds, and 0 otherwise. We further analyze the criterion of bucketization threshold selection in Appendix C. The bucketized Hadamard vectors $\widehat{\mathbf{H}}_Q$ and $\widehat{\mathbf{H}}_K$ are obtained by applying $B(\cdot)$ element-wise to $\mathbf{H}_Q$ and $\mathbf{H}_K$, respectively.

We further pack every eight elements into a single 16-bit value. The compressed $\widehat{\mathbf{H}}_K$ is stored directly in the KV cache, which increases cache size by only $1/16$, and is later used for lightweight similarity estimation. This strategy significantly reduces the memory footprint of the Hadamard-transformed vectors while retaining sufficient information for effective candidate token selection.

**Manhattan distance estimation.** The third part in Adamas is a similarity estimator based on Manhattan distance, operating directly on the 2-bit compressed representations. Given a compressed query $\widehat{\mathbf{H}}_Q \in \{0, 1, 2, 3\}^d$ and a compressed key $\widehat{\mathbf{H}}_K \in \{0, 1, 2, 3\}^d$, we approximate similarity by the negative Manhattan distance:

$$\text{sim}(\widehat{\mathbf{H}}_Q, \widehat{\mathbf{H}}_K) \approx -\|\widehat{\mathbf{H}}_Q - \widehat{\mathbf{H}}_K\|_1. \tag{6}$$

Since $\widehat{\mathbf{H}}_Q$ and $\widehat{\mathbf{H}}_K$ are encoded as 2-bit integers, the similarity computation can be carried out with bit-wise integer operations instead of relatively expensive floating-point arithmetic. To fully exploit this compression, we further design custom kernels that efficiently process groups of eight elements in each computation step, achieving both a low memory footprint and low computational overhead.

## 4 EXPERIMENTS

In this section, we evaluate Adamas in terms of both efficacy and efficiency. We also conduct ablation studies on the Hadamard transform, bucketization, and distance metrics to better understand the contribution of each component.

### 4.1 SETUP

**Models.** We use LongChat-v1.5-7b-32k (Li et al., 2023) and Yarn-Llama-2-7b-128k (Peng et al., 2024) for evaluation, following the experimental settings of Quest (Tang et al., 2024). LongChat-

Table 1: Average accuracy (%) of all methods evaluated on LongBench, normalized by full attention results.

| Methods | Adamas(Ours) | Adamas w/ L2 | Quest | SnapKV | StreamingLLM |
|---|---|---|---|---|---|
| Average Accuracy | 98.89 | 98.22 | 77.09 | 66.23 | 54.04 |

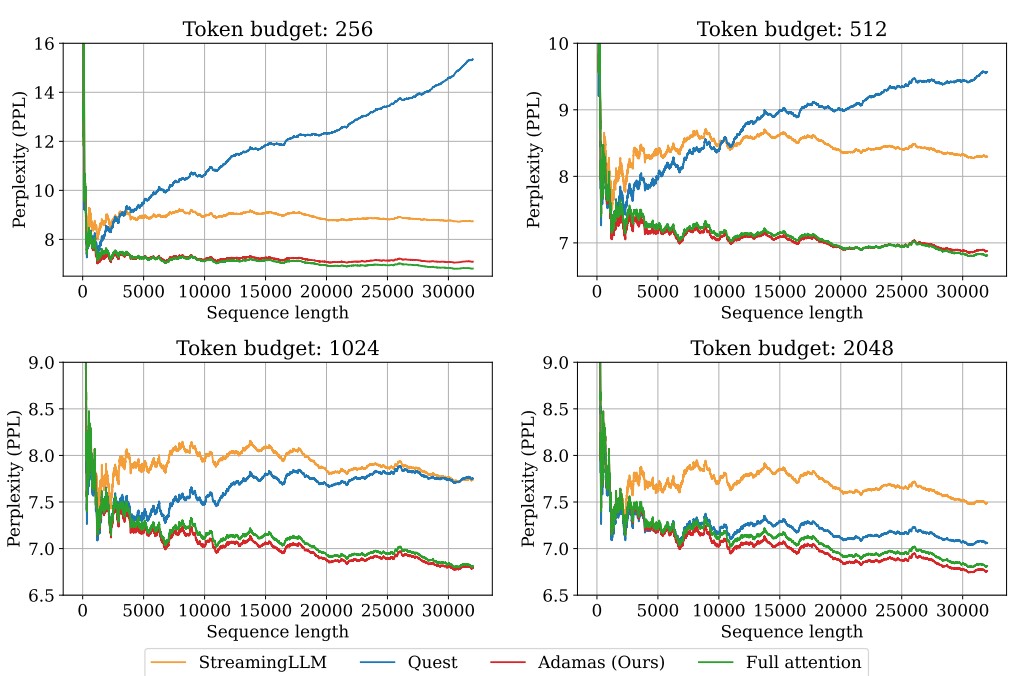

Figure 4: Perplexity results of StreamingLLM, Quest, Adamas, and full attention evaluated on PG19 with LongChat-7b-v1.5-32k under varying token budgets. Adamas consistently matches full attention and even shows lower perplexity at larger token budget.

v1.5-7b-32k is used in most experiments, while Yarn-Llama-2-7b-128k is employed for extremely long-context scenarios (up to 100K tokens).

**Tasks.** We consider three categories of evaluation tasks: (1) PG19 (Rae et al., 2019), a long-form language modeling dataset for assessing prediction confidence; (2) the passkey retrieval task (Peng et al., 2024), which measures retrieval capability in long contexts; and (3) six datasets from LongBench (Bai et al., 2024), a benchmark designed for long-context understanding. The Long-Bench tasks cover multiple settings, including single-document QA (NarrativeQA (Kočiský et al., 2018), Qasper (Dasigi et al., 2021), MultiFieldQA (Bai et al., 2024)), multi-document QA (Hot-potQA (Yang et al., 2018)), summarization (GovReport (Huang et al., 2021)), and few-shot learning (TriviaQA (Joshi et al., 2017)).

**Baselines.** For fair comparison, we benchmark against training-free sparse attention methods. Specifically, we include StreamingLLM (Xiao et al., 2023) as a representative static-sparse-mask method, SnapKV (Li et al., 2024) and Quest (Tang et al., 2024) as representative dynamic-KV-selection methods.

### 4.2 EFFICACY EVALUATION

**LongBench evaluation.** We evaluate Adamas and baselines on six datasets from LongBench. As shown in Figure 3, Adamas consistently surpasses these prior SOTA methods across all datasets, with particularly strong advantages under low token budgets, demonstrating its ability to preserve critical KV pairs. In comparison, StreamingLLM consistently underperforms full attention due to its KV cache eviction strategy. SnapKV exhibits mediocre overall accuracy and only achieves barely

Table 2: Passkey retrieval accuracy (%) of StreamingLLM, Quest, and Adamas under different token budgets. Results highlight the limitations of StreamingLLM's sliding window design, the budget sensitivity of Quest, and the robustness of Adamas across both short and long contexts.

(a) Results of 10K length passkey retrieval test on LongChat-7b-v1.5-32k

| Methods / Budget | 16 | 32 | 64 | 128 | 256 | 512 |
|---|---|---|---|---|---|---|
| StreamingLLM | 1% | 1% | 1% | 1% | 3% | 5% |
| Quest | 52% | 67% | **99%** | 98% | 100% | 100% |
| Adamas(Ours) | **68%** | **85%** | 93% | **98%** | **100%** | **100%** |

(b) Results of 100K length passkey retrieval test on Yarn-Llama-2-7b-128k

| Methods / Budget | 64 | 128 | 256 | 512 | 1024 | 2048 | 4096 |
|---|---|---|---|---|---|---|---|
| StreamingLLM | 1% | 1% | 1% | 1% | 1% | 2% | 4% |
| Quest | 25% | 58% | 84% | 95% | **99%** | 99% | 99% |
| Adamas(Ours) | **54%** | **71%** | **87%** | **95%** | 98% | **100%** | **100%** |

acceptable performance when the token budget exceeds 512, yet it still falls short of full attention. . Quest achieves competitive performance when the token budget exceeds 1024, but its accuracy drops sharply below 512. Specifically, as shown in Table 1, Adamas achieves over a 20% accuracy improvement over Quest, more than a 30% improvement over SnapKV, and accuracy comparable to Adamas with L2 on average accuracy values normalized by full attention results. For more detailed results, please refer to Table 4. These results confirm that Adamas sustains reliable accuracy across diverse long-context scenarios, even under constrained token budgets.

**Perplexity results.** We evaluate perplexity on the PG19 dataset using LongChat-7b-v1.5-32k with a 32K sequence length. As shown in Figure 4, Adamas closely tracks full attention across all settings and even achieves lower perplexity at larger token budgets, demonstrating its accuracy and robustness. In comparison, Quest suffers from high perplexity under high sparsity conditions, as its page-wise strategy introduces many irrelevant KV pairs, obscuring truly relevant ones when the token budget is limited. StreamingLLM maintains stable but consistently higher perplexity due to its eviction strategy, which permanently discards past information.

**Passkey results.** We evaluate passkey retrieval on LongChat-7b-v1.5-32k with 10K-length inputs and on Yarn-Llama-2-7b-128k with 100K-length inputs. Results are listed in Table 2. Adamas maintains consistently high accuracy across all budgets and significantly outperforms prior SOTA under constrained settings. In comparison, StreamingLLM performs poorly across all budgets due to its sliding window design, which discards keys outside the retained span. Quest achieves strong accuracy with large budgets but degrades sharply when the budget is small ($\leq 32$ for 10K and $\leq 128$ for 100K), as its page-wise partitioning fails to recall scattered keys. Overall, these results highlight that Adamas robustly retrieves critical KV pairs even under severely limited token budgets.

### 4.3 EFFICIENCY EVALUATION

We conduct efficiency evaluations with LongChat-7b-v1.5-32k on NVIDIA RTX A6000.

**Kernels.** We implement high-performance CUDA kernels for the Hadamard transform, fused bucketization–compression, Manhattan-distance estimation, top-$k$ selection, and sparse attention. Manhattan-distance estimation, the most computationally intensive component (shown in Appendix Table 8), is significantly accelerated through lightweight bit-wise integer operations enabled by our bucketization and compression design. The Hadamard transform and fused bucketization-compression incur negligible overhead and are thus omitted from further analysis. As shown in Figure 5, Adamas achieves consistent acceleration across various token budgets and sequence lengths. In particular, it delivers up to $4.4\times$ speedup over full attention implemented by FlashInfer (Ye et al., 2025) under a 256-token budget, with nearly lossless accuracy.

**End-to-end.** Table 3 shows that as the token budget increases from 256 to 4096, Adamas maintains stable end-to-end latency and achieves up to $1.5\times$ speedup over FlashInfer on 32K-length

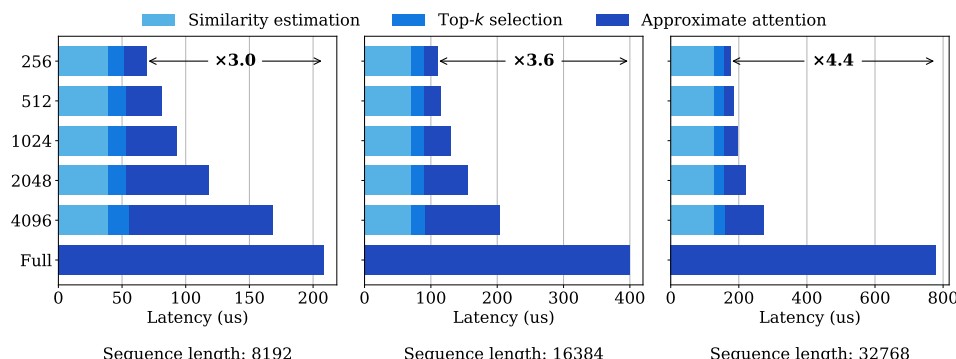

Figure 5: Kernel-level breakdown of Adamas self-attention compared to full attention (FlashInfer).

Table 3: End-to-end decoding latency (ms). Adamas achieves up to $1.5\times$ speedup.

| Sequence / Budget | 256 | 512 | 1024 | 2048 | 4096 | Full Attn |
|---|---|---|---|---|---|---|
| 8192 | 26.0 | 26.3 | 26.7 | 27.5 | 28.8 | 30.2 |
| 16384 | 29.3 | 29.5 | 30.0 | 30.9 | 32.3 | 38.0 |
| 32768 | 34.9 | 35.2 | 35.8 | 36.9 | 38.6 | 53.7 |

sequences. Since the attention module accounts for only $30\% - 40\%$ of the total decoding time, the overall acceleration is somewhat diluted compared to the kernel-level results. Nevertheless, Adamas exhibits only marginal increases in latency with larger token budgets, highlighting its scalability and efficiency in long-context scenarios. We further conduct end-to-end efficiency evaluations on an A800 80G PCIe GPU, which better reflects real-world server environments. Notably, Adamas achieves up to a 1.4× speedup on 32K-length sequences, demonstrating substantial and practically meaningful efficiency gains. Please refer to Appendix B for more detailed results.

## 4.4 ABLATION STUDY

In this part, we conduct ablation studies on the Hadamard transform, bucketization, and distance metrics. To better understand the contribution of each component, we benchmark three variants of Adamas on LongBench using LongChat-7b-v1.5-32k: (1) Adamas without the Hadamard transform, (2) Adamas with different bucketization granularities, and (3) Adamas with L2 distance (replacing Manhattan distance). The results are shown in Figure 6, with detailed data provided in Appendix H.

**Hadamard transform.** As illustrated in Figure 6, Adamas without the Hadamard transform (Adamas w/o Hadamard) directly bucketizes queries and keys before Manhattan-distance estimation, achieving near-zero scores across multiple datasets under small token budgets. Its performance improves only gradually with larger budgets and remains below the baseline even at a 4096-token budget. This behavior indicates that the model fails to generate accurate answers without the Hadamard transform, underscoring its critical role in Adamas: Hadamard transform smooths the value distribution of queries and keys, suppresses extreme outliers, and thereby mitigates the information loss introduced by bucketization.

**Bucketization.** To investigate how bucketization granularity affects accuracy, we compare three variants: 1-bit (Adamas-1bit, with threshlod), 2-bit (Adamas), and 3-bit (Adamas-3bit). As shown in Figure 6, Adamas and Adamas-3bit achieve nearly identical performance, with Adamas-3bit showing only a slight advantage. Adamas-1bit performs comparably to Adamas when the token budget exceeds 256, but its accuracy drops significantly under smaller budgets, suggesting that retaining more bits helps preserve information. However, the marginal improvement of Adamas-3bit over Adamas highlights diminishing returns when increasing bit width. Since higher bit widths also incur greater storage costs, we adopt 2-bit bucketization as the most memory-efficient choice without sacrificing accuracy.

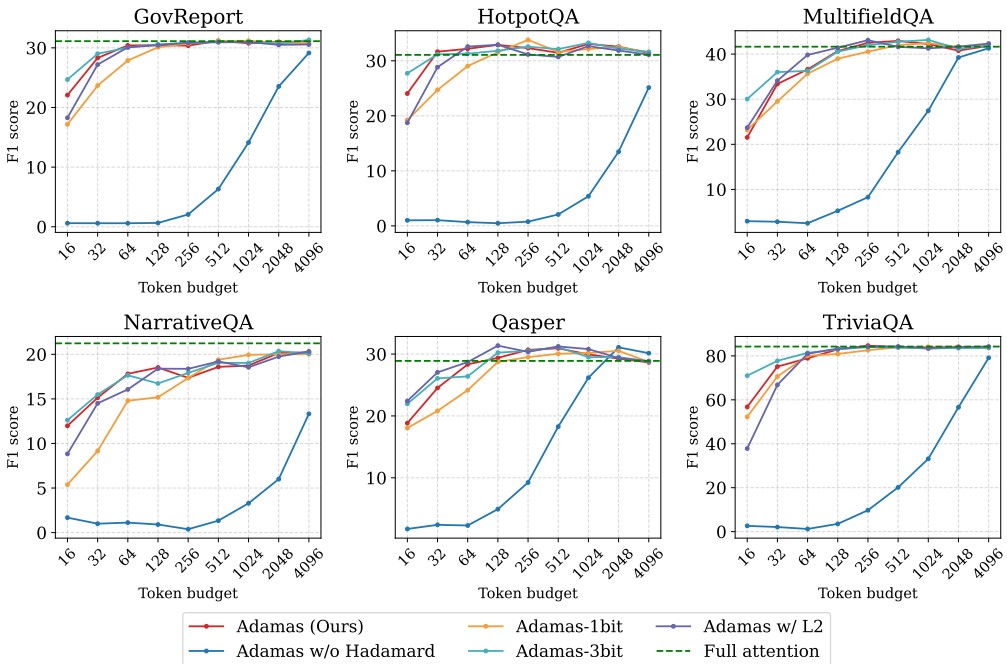

Figure 6: Ablation studies on the Hadamard transform, bucketization, and distance metrics.

**Distance metrics.** We replace Manhattan distance (L1) estimator with L2 distance estimator to assess the impact of distance metrics. As shown in Figure 6, Adamas with L2 distance achieves comparable performance with vanilla Adamas, showing similar performance curves. Specifically, Adamas with L2 performs slightly better on single-document QA such as MultifieldQA and Qasper, while vanilla Adamas shows margin advantages on other tasks. This discrepancy reflects the different sensitivities of the two distance metrics: L2 distance emphasizes global geometric consistency and is more effective for reasoning within a single coherent context, whereas L1 distance is more robust to noise and sparsity, making it better suited for integrating dispersed or partially relevant information across multiple sources.

## 5 RELATED WORK

A large amount of work has explored sparsity in attention mechanisms to improve computational efficiency, with a primary focus on how to effectively select critical KV pairs. Existing methods can be broadly categorized into three groups:

**Static sparse methods.** These methods adopt predefined sparsity patterns, such as sliding windows or attention sinks. For example, H2O (Zhang et al., 2023) and infLLM (Xiao et al., 2024) rely on the sliding window pattern, while StreamingLLM (Xiao et al., 2023) and DUOAttention (Xiao et al., 2025) combine sliding window with attention sink to preserve accuracy. Moreover, DUOAttention requires an extra training phase to determine streaming/retrieval heads, which introduces substantial training overhead. MOA (Wang et al., 2024a) further refines this idea by profiling each attention head across layers to adjust the window span and then incorporating attention sinks to build the sparse mask. Although these static designs yield high throughput, their fixed structures risk discarding tokens that are critical to a given query (Tang et al., 2024), often resulting in accuracy degradation when the token budget is constrained, therefore significantly suffering from retrieval tasks.

**Dynamic KV selection methods.** To better preserve accuracy, these approaches select KV pairs adaptively based on the input query. Quest (Tang et al., 2024) estimates page importance using an upper-bound approximation of attention weights and selects the top-$k$ pages for sparse attention, though its page-level granularity may miss critical tokens. MInference (Jiang et al., 2024) defines three dynamic patterns and performs online estimation per attention head to select the proper pattern,

while FlexPrefill (Lai et al., 2025) adaptively determines sparse patterns and selection ratios per head to accelerate long-sequence prefilling. Both methods, however, primarily target the prefilling stage rather than full-sequence decoding , therefore are orthogonal and can, in principle, be combined with Adamas. SeerAttention (Gao et al., 2024) introduces a learnable gating mechanism that activates important blocks within the attention map, but at the cost of training additional parameters, which complicates deployment by preventing the model from being used out-of-the-box at inference.

**Training-based sparse methods.** Beyond static and dynamic masking, an alternative approach integrates sparsity into the attention mechanism through specialized training procedures. Reformer (Kitaev et al., 2020) replaces standard dot-product attention with locality-sensitive hashing (LSH), reducing complexity from $O(L^2)$ to $O(L \log L)$. NSA (Yuan et al., 2025) adopts a dynamic hierarchical sparse strategy that combines coarse-grained token compression, fine-grained token selection, and sliding window via gating to balance global context and local precision. MoBA (Lu et al., 2025) incorporates a Mixture-of-Experts (MoE) design (Shazeer et al., 2017) into sparse attention, using a gating mechanism to select historical blocks. While these methods often achieve better accuracy and efficiency than standard full attention, they require pretraining the model, which is computationally costly and limits their applicability. In contrast, our method is training-free and can be directly applied to pretrained LLMs, offering broad compatibility without retraining overhead.

In parallel to sparse attention, another line of research focuses on **KV cache compression**, which aims to reduce the memory footprint of stored key/value tensors. While decoding with a KV cache has a baseline time and space complexity of $O(T)$ with respect to sequence length $T$, these two families of techniques address different bottlenecks. Sparse attention focuses on selecting the most relevant query-key pairs at the attention level, with the overall goal of approximating full attention computation using a subset of tokens to achieve acceleration. This reduces I/O during attention computation, especially under high sparsity, and thus primarily improves latency (i.e., reduces time complexity), rather than memory usage. In contrast, KV cache compression reduces the memory footprint of stored key/value tensors (space complexity). However, because compression or pruning of tokens is generally irreversible, such methods often incur accuracy loss and may introduce additional decoding latency. Moreover, simply compressing the KV cache alone may not necessarily reduce I/O during attention computation, limiting its effectiveness for speedup. Overall, sparse attention and KV cache compression are largely complementary: the former focuses on computational acceleration, while the latter focuses on memory efficiency. Each addresses a different bottleneck in efficient long-context decoding.

## 6 CONCLUSION

In this paper, we introduce Adamas, a lightweight yet efficient attention mechanism for long-context inference. By integrating the Hadamard transform, bucketization, 2-bit compression, and a Manhattan-distance estimator, Adamas dynamically selects the most relevant KV pairs for each query. This design enables Adamas to achieve accuracy comparable to full attention with only a 64-token budget, and to become nearly lossless at 128 tokens, attaining up to $8\times$ higher sparsity than prior SOTA methods. Comprehensive evaluations demonstrate that Adamas delivers up to $4.4\times$ self-attention and $1.5\times$ end-to-end speedups on 32K-length sequences, while achieving perplexity even lower than full attention and surpassing prior SOTA by more than $2\times$.

## REPRODUCIBILITY STATEMENT

To ensure reproducibility, we provide the complete implementation of Adamas in the anonymized repository, including algorithmic details, source code and scripts for running experiments. All datasets and models used in this work are publicly available. We are confident that with the provided resources, readers can reproduce the entirety of our presented results.

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

## A  LONGBENCH EVALUATIONS

We present detailed data of Figure 3 in Table 4 for reference. As shown in Table 4, both variants of Adamas (vanilla Adamas and Adamas with L2 distance) show consistent better performance over other sparse methods, especially under limited token budget (less than 512 tokens)

Table 4: Detailed LongBench evaluation results of StreamingLLM, SnapKV, Quest, Adamas with L2 and Adamas(Ours) on LongChat-7b-v1.5-32k.

| Datasets | Methods / Budget | 64 | 128 | 256 | 512 | 1024 | 2048 | 4096 |
|---|---|---|---|---|---|---|---|---|
| GovReport (31.12) | StreamingLLM | 1.21 | 7.13 | 14.52 | 18.86 | 21.72 | 24.08 | 27.01 |
| | SnapKV | 1.55 | 1.21 | 3.66 | 21.11 | 23.62 | 26.31 | 28.47 |
| | Quest | 3.3 | 11.93 | 22.84 | 27.27 | 29.89 | **31.19** | **31.23** |
| | Adamas w/ L2 | **30.09** | **30.45** | **30.86** | 30.93 | **30.96** | 30.50 | 30.56 |
| | Adamas(Ours) | **30.41** | **30.44** | 30.37 | **31.18** | 30.77 | **31.00** | **31.07** |
| HotpotQA (31.07) | StreamingLLM | 3.63 | 9.68 | 14.51 | 16.25 | 18.62 | 21.83 | 24.14 |
| | SnapKV | 10.55 | 12.16 | 20.28 | **35.65** | 33.90 | 35.23 | 33.94 |
| | Quest | 5.31 | 15.49 | 21.54 | 26.64 | 30.29 | 32.21 | **32.93** |
| | Adamas w/ L2 | **32.58** | **32.91** | 31.15 | 30.70 | 32.66 | 31.86 | 31.10 |
| | Adamas(Ours) | **32.18** | **32.89** | **32.30** | **31.45** | **33.06** | **32.56** | 31.41 |
| MultifieldQA (41.64) | StreamingLLM | 2.70 | 11.67 | 17.93 | 20.01 | 21.62 | 27.07 | 34.21 |
| | SnapKV | 7.53 | 6.24 | 16.74 | 40.98 | 37.76 | 40.03 | **42.82** |
| | Quest | 6.99 | 20.87 | 31.12 | 39.8 | **42.03** | **44.09** | 43.25 |
| | Adamas w/ L2 | **39.81** | **41.41** | **43.08** | 41.60 | 41.28 | **41.65** | 42.33 |
| | Adamas(Ours) | **36.60** | **40.67** | **42.49** | **42.93** | 42.22 | 40.77 | 41.83 |
| NarrativeQA (21.23) | StreamingLLM | 0.92 | 4.47 | 9.17 | 10.61 | 12.46 | 16.85 | 17.40 |
| | SnapKV | 4.62 | 2.45 | 9.5 | 17.97 | **19.47** | 19.31 | **20.32** |
| | Quest | 1.86 | 5.98 | 14.2 | 16.28 | 17.91 | 19.66 | 19.88 |
| | Adamas w/ L2 | **16.06** | **18.39** | **18.37** | **19.20** | 18.54 | **19.74** | **20.33** |
| | Adamas(Ours) | **17.81** | **18.52** | 17.36 | **18.59** | **18.74** | **20.14** | 20.22 |
| Qasper (28.89) | StreamingLLM | 3.15 | 8.72 | 10.87 | 11.74 | 14.79 | 17.50 | 25.33 |
| | SnapKV | 11.11 | 8.97 | 9.02 | 25.25 | 26.40 | 28.92 | **29.25** |
| | Quest | 8.67 | 18.54 | 26.72 | 30.89 | **31.01** | **31.86** | 29.79 |
| | Adamas w/ L2 | **28.70** | **31.36** | 30.35 | **31.23** | 30.80 | 29.36 | 28.86 |
| | Adamas(Ours) | **28.33** | **29.38** | **30.68** | **30.93** | 29.94 | 29.26 | 28.65 |
| TriviaQA (84.25) | StreamingLLM | 8.73 | 33.14 | 51.20 | 60.37 | 68.48 | 75.67 | 80.21 |
| | SnapKV | 0.67 | 9.19 | 55.92 | **84.01** | 80.88 | 80.12 | 81.25 |
| | Quest | 12.29 | 44.68 | 68.92 | 81.32 | **83.95** | **85.84** | **84.94** |
| | Adamas w/ L2 | **80.97** | **83.35** | 84.21 | 84.29 | 83.60 | 84.03 | **84.25** |
| | Adamas(Ours) | **78.91** | **82.95** | **84.67** | 83.99 | 83.36 | 83.75 | 83.95 |

## B  END-TO-END EFFICIENCY EVALUATIONS

We further conduct end-to-end efficiency evaluations on an A800 80G PCIe GPU, which better reflects real-world server environments. As shown in Table 5, Adamas achieves up to a $1.4\times$ speedup on 32K-length sequences, demonstrating substantial and practically meaningful efficiency gains.

It is worth noting that Quest employs page-wise sparsity, trading accuracy for speed, especially under high-sparsity regimes, whereas Adamas consistently maintains high accuracy, achieving over 20% accuracy gains over Quest on average (Table 1. Despite operating at the token-wise level, Adamas

Table 5: End-to-end efficiency(s) evaluated on an A800 80G PCIe GPU, encompassing both prefill and 64-token decoding stages. Adamas achieves up to $1.4\times$ speedup with high accuracy.

| Sequence | Methods / Budget | 256 | 512 | 1024 | 2048 | 4096 | Full Attn |
|---|---|---|---|---|---|---|---|
| 8192 | Adamas(Ours) | 1.833 | 1.863 | 1.844 | 1.751 | 1.785 | |
| | Quest | 1.626 | 1.606 | 1.573 | 1.596 | 1.621 | 2.343 |
| | SnapKV | 2.373 | 2.339 | 2.351 | 2.315 | 2.331 | |
| 16384 | Adamas(Ours) | 2.828 | 2.834 | 2.861 | 2.894 | 2.924 | |
| | Quest | 2.473 | 2.536 | 2.503 | 2.492 | 2.512 | 3.930 |
| | SnapKV | 3.851 | 3.934 | 3.901 | 3.938 | 3.914 | |
| 32768 | Adamas(Ours) | 5.680 | 5.711 | 5.703 | 5.730 | 5.804 | |
| | Quest | 4.901 | 4.908 | 4.918 | 4.945 | 4.984 | 7.978 |
| | SnapKV | 7.966 | 7.945 | 7.953 | 7.976 | 7.967 | |

is only about 10% slower than Quest, underscoring the effectiveness of our carefully optimized high-performance kernel.

## C  BUCKETIZATION THRESHOLD ANALYSIS

We analyzed the distributions of Queries and Keys using the LongBench dataset, with the results presented in Figure 10. The distributions indicate that both Queries and Keys closely follow a zero-mean normal distribution.

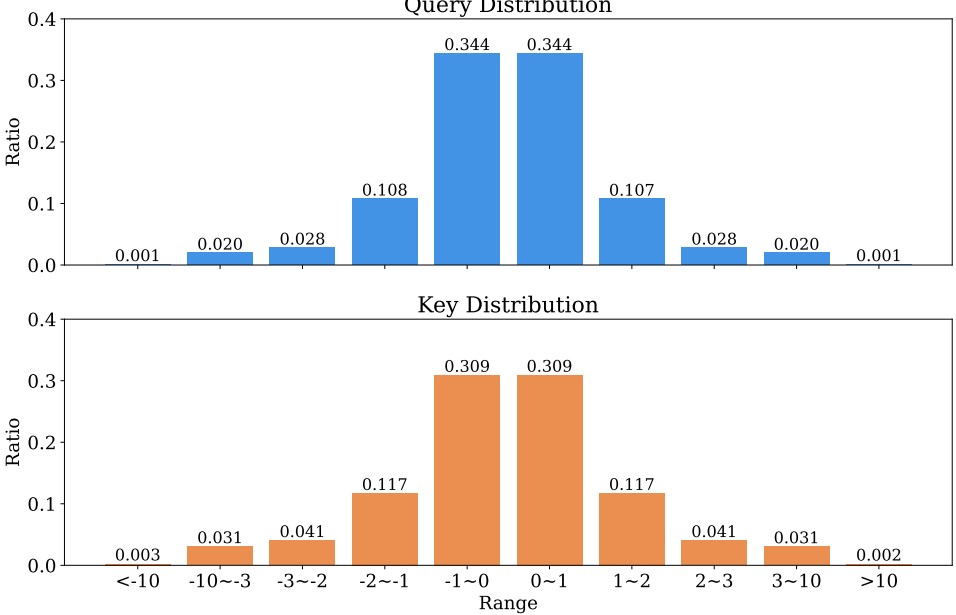

Figure 7: Distribution of Queries and Keys derived from LongBench statistics. Both Queries and Keys exhibit an approximately normal distribution.

To study the sensitivity of bucketization thresholds, we evaluate a range of values $b \in \{1, 3, 5, 7, 10, 15\}$ under a fixed token budget of 128 using the bucketization rule $[-b, 0, b]$. The results in Table 6, evaluated on six LongBench datasets (with top-2 scores bolded), show that $b = 1$ and $b = 10$ consistently achieve the best accuracy across tasks.

Table 6: Sensitivity study of bucketization threshold, evalueted on six datasets of LongBench. Top-2 scores are bolded. Both threshold=1 and threshold=10 show consistent better accuracy performance.

| Dataset / Threshold | 1 | 3 | 5 | 7 | 10 | 15 |
|---|---|---|---|---|---|---|
| GovReport | **31.06** | 25.29 | 29.99 | 29.62 | **30.44** | 29.20 |
| HotpotQA | **33.38** | 26.12 | 30.81 | 29.00 | **32.89** | 31.54 |
| MultifieldQA | 40.28 | 33.54 | 37.75 | **41.65** | **40.67** | 39.35 |
| NarrativeQA | **17.96** | 16.47 | 15.45 | 16.02 | **18.52** | 14.90 |
| Qasper | **28.95** | 23.47 | 27.15 | 28.16 | **29.38** | **29.38** |
| TriviaQA | **83.39** | 74.7 | 80.46 | 81.04 | **82.95** | 80.65 |

## C.1 EMPIRICAL SENSITIVITY RESULTS

The experimental results reveal two threshold regimes that yield notably strong performance:

- **Fine-grained threshold:** $b = 1$ (in implicit standardized units by LayerNorm)
- **Coarse-grained threshold:** $b = 10$ (in raw magnitude)

Intermediate thresholds (e.g., $b = 3, 5, 7$) and those exceeding $b = 10$ exhibit moderate degradation but remain functional, indicating robustness to suboptimal threshold selection.

## C.2 COMPREHENSIVE LONGBENCH COMPARISON FOR $b = 1$ AND $b = 10$

To further validate the effectiveness of the two best-performing thresholds, we conduct comprehensive evaluations of $b = 1$ and $b = 10$ across the LongBench benchmark. The results, presented in Figure 8, show that the performance curves of $b = 1$ and $b = 10$ almost completely overlap on every task. Moreover, both thresholds significantly outperform all other baselines, demonstrating that these two bucketization schemes consistently preserve the essential features required for accurate QK ranking under sparse attention.

## C.3 WHY $b = 1$ AND $b = 10$ PERFORM WELL

The effectiveness of these two thresholds can be explained by how they isolate dominant contributors to the QK dot product:

- **Large-magnitude components** ($|x| > 10$) disproportionately influence the QK dot product and often encode strong semantic similarity.
- **Near-zero components** ($|x| < 1$) contribute minimally and behave largely as noise.

Under this interpretation:

- $b = 1$ provides a principled separation of "strong positive," "strong negative," and "near-zero" signals in standardized units, enabling a clean distinction between informative and uninformative dimensions.
- $b = 10$ emphasizes dimensions with high raw magnitude that dominate the attention computation.

Both thresholds thus preserve the structural information necessary for reliably approximating QK rankings.

## C.4 WHY INTERMEDIATE OR LARGER THRESHOLDS UNDERPERFORM

Thresholds in the mid-range or in overly coarse regimes may impair bucketization by:

- **Over-partitioning** weak-signal regions, thereby amplifying noise, or
- **Over-merging** strong-signal regions, diminishing contrast among informative dimensions.

These effects lead to mild degradation in similarity estimation and consequently reduced accuracy.

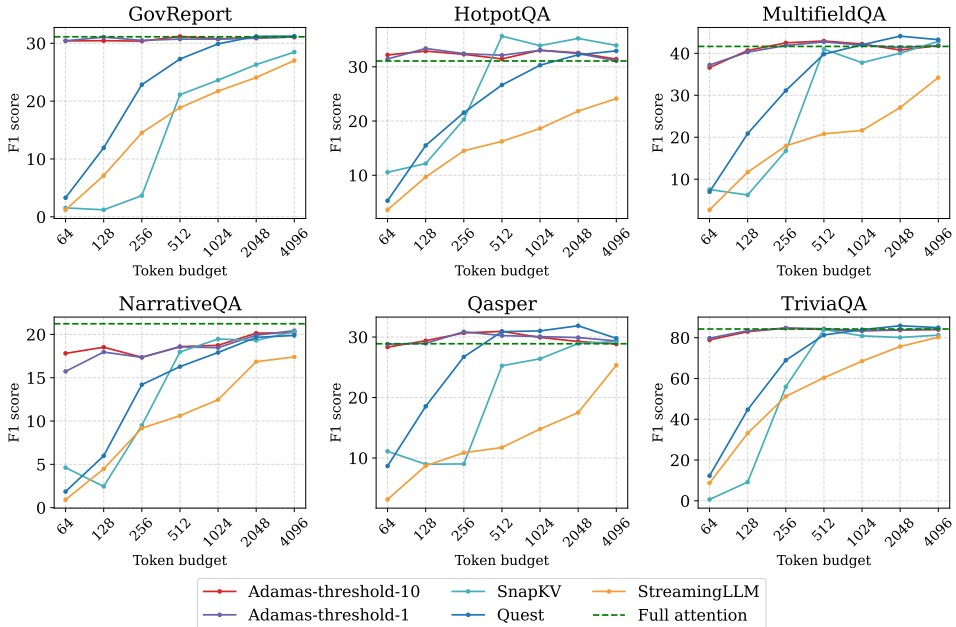

Figure 8: Full LongBench comparison of bucketization thresholds $b = 1$ and $b = 10$. The accuracy curves nearly coincide and significantly outperform other baselines, validating the effectiveness of both threshold choices.

### C.5 CONCLUSION

Given that the dimensions of Queries and Keys approximately follow a normal distribution, an effective threshold need only separate strong-contribution features from weak or noisy ones. Thresholds that fail to achieve this separation exhibit moderate performance loss. In contrast, $b = 1$ and $b = 10$ provide interpretable and complementary partitioning strategies that retain the essential information needed for robust approximation of the QK ranking. These findings establish both thresholds as sound, empirically validated choices for bucketized attention computation.

## D PERPLEXITY EVALUATION OF ADAMAS WITH 4096-TOKEN BUDGET

For completeness, we additionally include the perplexity curves under a 4096-token budget, evaluated on PG19. As shown in the Figure 9, Adamas achieves noticeably lower perplexity than full attention, indicating that it can effectively filter out noisy tokens, select the most critical ones, and perform more accurate inference.

## E NEEDLE-IN-A-HAYSTACK STRESS TESTS

We conduct the Needle-in-a-Haystack stress tests using Yarn-Llama-2-7b-128k model, to evaluate the recall rates of Adamas, Quest, and full attention at a context length of 32K. As shown in Figure 10 and 11, Adamas maintains a recall performance that is nearly identical to that of full attention even under high sparsity (with only 256-token budget), consistently outperforming the baseline. Specifically, both Adamas and full attention achieve scores around 7/10 at a sequence length of 32K, whereas Quest reaches only a little over 2. This finding robustly supports our claim of "better recall under high sparsity".

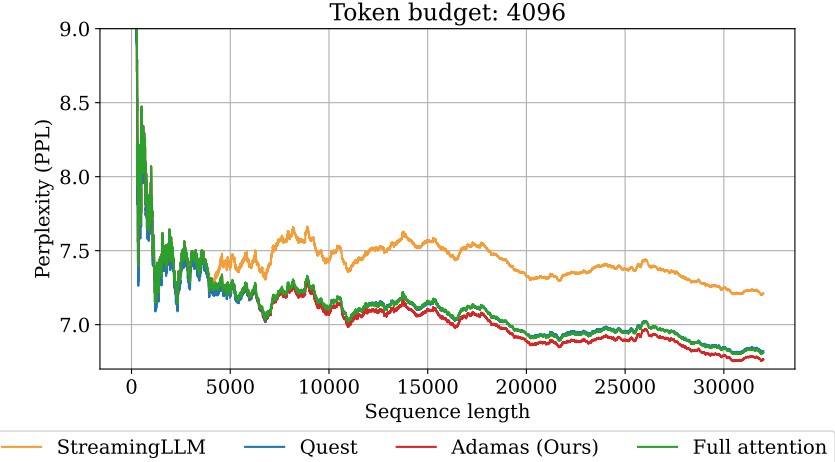

Figure 9: Perplexity comparison on PG19 under a 4096-token budget. Adamas achieves noticeably lower perplexity than full attention.

## F LLM USAGE

In this work, we used LLMs as auxiliary tools for writing and figure preparation. Specifically, LLMs were employed to refine the clarity of text, improve grammar, and correct minor wording errors. In addition, for some figures, initial plotting scripts were generated with the help of LLMs, which we subsequently modified and finalized. Importantly, all research ideas, theoretical results, algorithm designs, and experimental analyses were developed independently by the authors without reliance on LLMs.

## G COMPUTATIONAL WORKLOADS AND MEMORY ACCESS ANALYSIS

### G.1 MATHEMATICAL NOTATIONS

Table 7: Mathematical notations.

| Symbol | Description. |
|---:|---|
| $b$ | Batch size |
| $s$ | Sequence length |
| $h$ | Hidden dimension |
| $h_{kv}$ | Hidden dimension of Key and Value |
| $d$ | Head dimension |
| $n$ | Number of buckets |
| $p$ | Page size |
| $k$ | Number of selected tokens |

### G.2 COMPUTATIONAL WORKLOAD

Detailed breakdown of computational workload is shown in Table 8. In summary:

- Computational workload of full attention: $4(h + h_{kv} + s)bh$ FLOPs.

- Computational workload of Quest: $\left[4h + 4h_{kv} + \left(\frac{4}{p} + \frac{4log_2(\frac{k}{p})}{pd}\right)s + 4k\right] \cdot bh + 2ph$ FLOPs.

- Computational workload of Adamas: $\left[4h + 4h_{kv} + 2n + 2 + \left(\frac{4log_2(k)}{d} + 3\right)s + 4k\right] \cdot bh$ FLOPs.

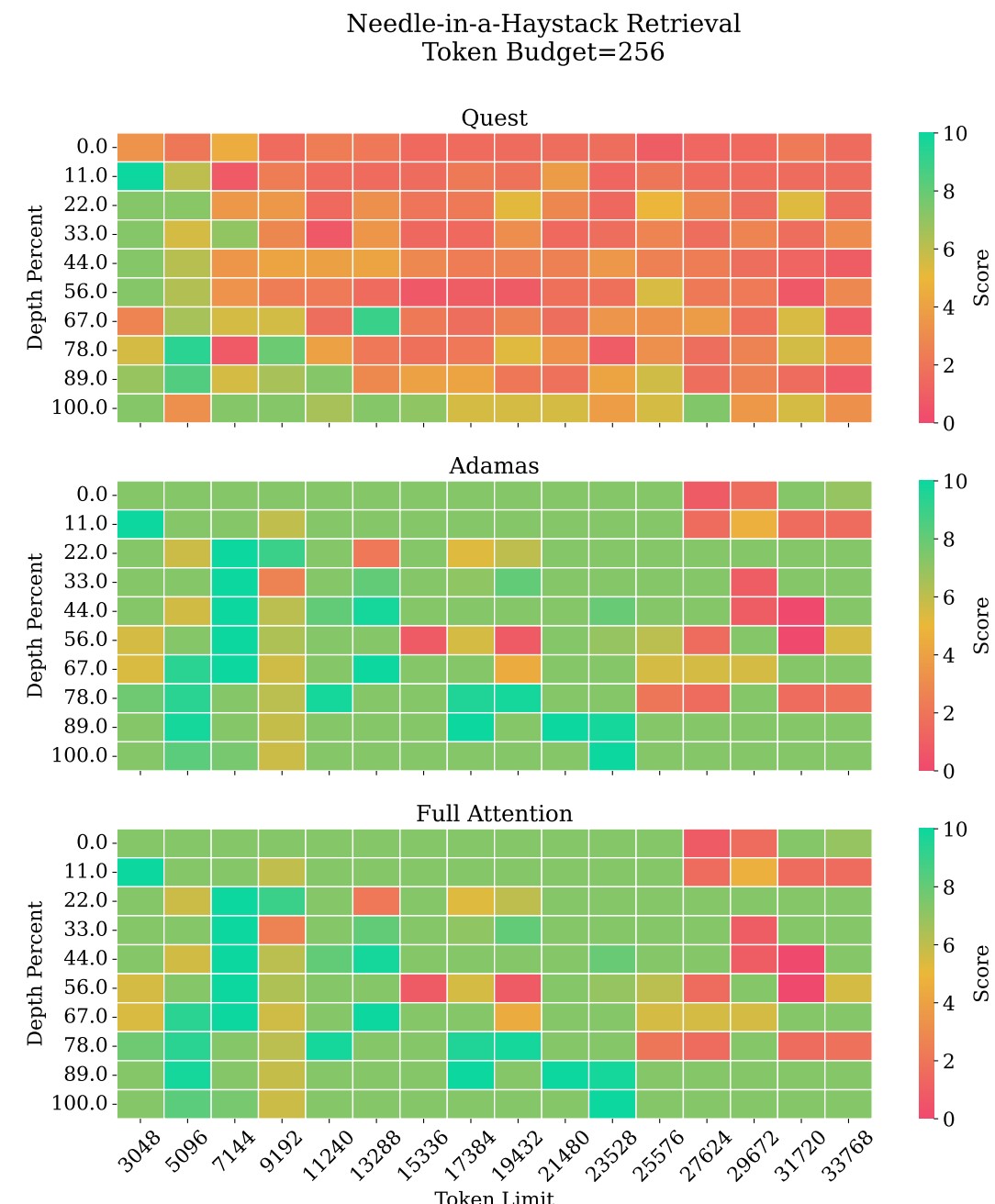

Figure 10: Needle-in-a-Haystack stress test using Yarn-Llama-2-7b-128k, with token budget=256.

### G.3 MEMORY ACCESS

Detailed breakdown of memory access is shown in Table 9. In summary:

- Memory access of full attention:
  - Read access: $3bh + 2bsh_{kv} + 2h^2 + 2hh_{kv} + h + h_{kv}$
  - Write access: $3bh + 2bh_{kv}$
- Memory access of Quest:

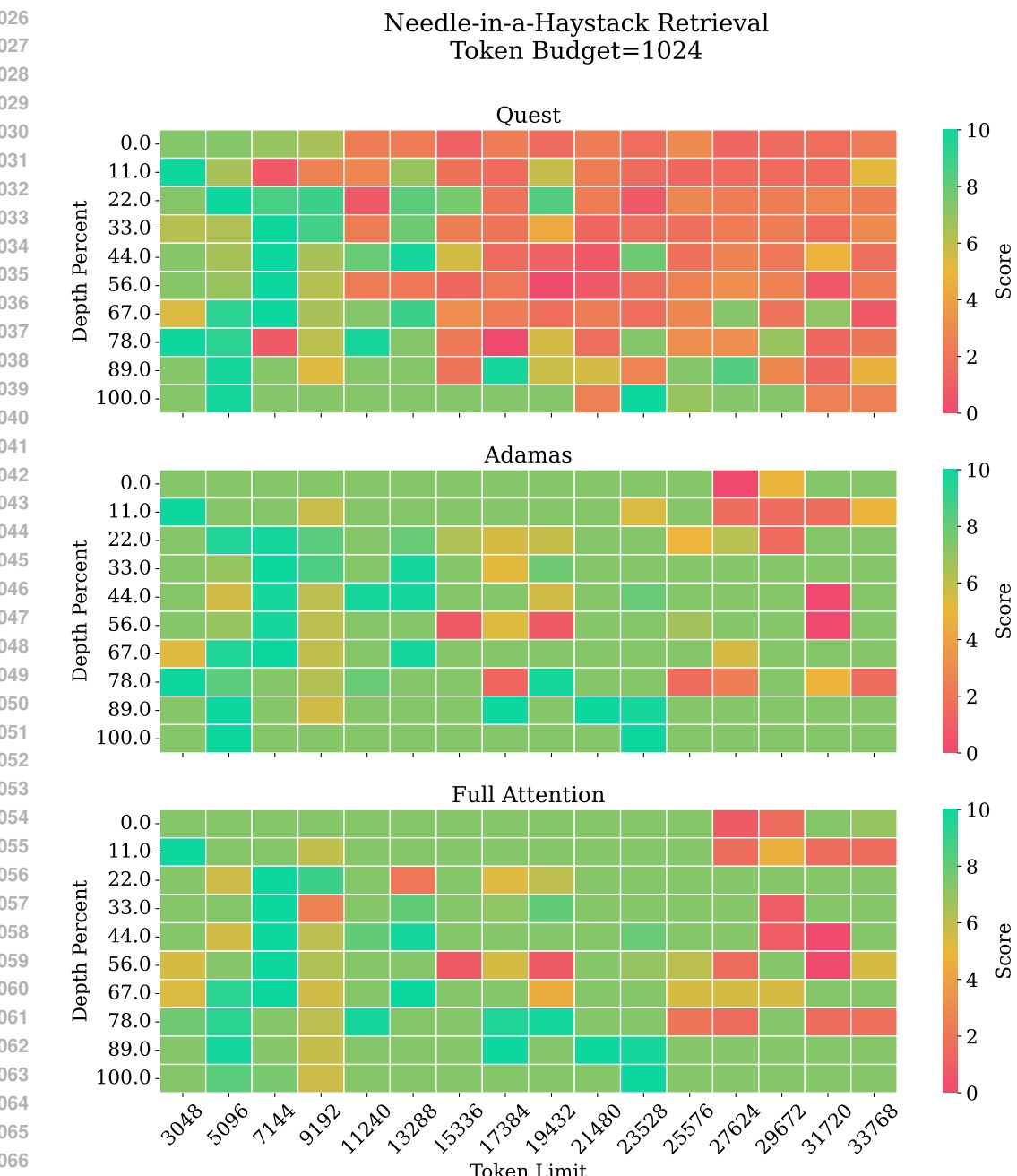

Figure 11: Needle-in-a-Haystack stress test using Yarn-Llama-2-7b-128k, with token budget=1024.

- Read access: $7bh + \frac{2d+1}{pd} \cdot bsh + 2bkh_{kv} + 2h^2 + 2hh_{kv} + h + 2h_{kv}$

- Write access: $(5 + \frac{s+k}{pd})bh + 2bh_{kv}$

- Memory access of Adamas:

  - Read access: $\frac{57}{8}bh + \frac{bsh}{d} + 2bkh_{kv} + 2d^2 + 2h^2 + 2hh_{kv} + h + 2h_{kv}$

  - Write access: $(\frac{21}{4} + \frac{s+k}{d})bh + 2bh_{kv}$

Table 8: Computational workload

| | Component | FLOPs |
|---|---|---|
| Full attention | QKV projection | $2bh(h + 2h_{kv})$ |
| | $P = \text{softmax}(QK^\top)$ | $2bsh$ |
| | $PV$ | $2bsh$ |
| | Output projection | $2bh^2$ |
| Quest | QKV projection | $2bh(h + 2h_{kv})$ |
| | Reduce keys | $2ph$ |
| | QK element-wise product | $\frac{2}{p} \cdot bsh$ |
| | Per channel max | $\frac{1}{p} \cdot bsh$ |
| | Page sum | $\frac{1}{p} \cdot bsh$ |
| | Top-$k$ | $\frac{4log_2(\frac{k}{p})}{pd} \cdot bsh$ |
| | $P = \text{softmax}(QK^\top)$ | $2bkh$ |
| | $PV$ | $2bkh$ |
| | Output projection | $2bh^2$ |
| Adamas | QKV projection | $2bh(h + 2h_{kv})$ |
| | Hadamard transform | $2bh$ |
| | Bucketizaion | $2n \cdot bh$ |
| | Manhattan-distance estimation | $3bsh$ |
| | Top-$k$ | $\frac{4log_2(k)}{d} \cdot bsh$ |
| | $P = \text{softmax}(QK^\top)$ | $2bkh$ |
| | $PV$ | $2bkh$ |
| | Output projection | $2bh^2$ |

Table 9: Memory access

| | Component | Read access | Write access |
|---|---|---|---|
| Full attention | QKV projection | $bh + h(h + 2h_{kv}) + (h + 2h_{kv})$ | $b(h + 2h_{kv})$ |
| | $A = \text{softmax}(QK^\top)V$ | $bh + 2bsh_{kv}$ | $bh$ |
| | Output projection | $bh + h^2$ | $bh$ |
| Quest | QKV projection | $bh + h(h + 2h_{kv}) + (h + 2h_{kv})$ | $b(h + 2h_{kv})$ |
| | Reduce keys | $3bh$ | $2ph$ |
| | Criticality estimation | $\frac{2bsh}{p} + bh$ | $\frac{bsh}{pd}$ |
| | Top-$k$ | $\frac{bsh}{pd}$ | $\frac{bkh}{pd}$ |
| | $A = \text{softmax}(QK^\top)V$ | $bh + 2bsh_{kv}$ | $bh$ |
| | Output projection | $bh + h^2$ | $bh$ |
| Adamas | QKV projection | $bh + h(h + 2h_{kv}) + (h + 2h_{kv})$ | $b(h + 2h_{kv})$ |
| | Hadamard transform | $2bh + 2hd^2$ | $2bh$ |
| | Bucketizaion | $2bh$ | $\frac{bh}{4}$ |
| | Manhattan-distance estimation | $\frac{bh+bsh}{8}$ | $\frac{bsh}{d}$ |
| | Top-$k$ | $\frac{bsh}{d}$ | $\frac{bkh}{d}$ |
| | $A = \text{softmax}(QK^\top)V$ | $bh + 2bsh_{kv}$ | $bh$ |
| | Output projection | $bh + h^2$ | $bh + h^2$ |

# H  ABLATION STUDIES

Below we present detailed ablation results evaluated on LongBench with LongChat-7b-v1.5-32k in Table H.

Table 10: Ablation results evaluated on LongBench with LongChat-7b-v1.5-32k.

| Datasets | Methods / Budget | 16 | 32 | 64 | 128 | 256 | 512 | 1024 | 2048 | 4096 |
|---|---|---|---|---|---|---|---|---|---|---|
| GovReport (31.12) | Adamas (Ours) | 22.08 | 28.31 | **30.41** | 30.44 | 30.37 | **31.18** | 30.77 | **31.00** | 31.07 |
| | Adamas w/o Hadamard | 0.60 | 0.59 | 0.59 | 0.64 | 2.06 | 6.32 | 14.12 | 23.53 | 29.14 |
| | Adamas-1bit | 17.17 | 23.67 | 27.88 | 30.09 | 30.72 | 31.16 | **31.38** | 30.84 | 30.70 |
| | Adamas-3bit | **24.69** | **29.01** | 30.05 | **30.59** | 30.83 | 31.01 | 30.95 | 30.71 | **31.33** |
| | Adamas w/ L2 distance | 18.27 | 27.19 | 30.09 | 30.45 | **30.86** | 30.93 | 30.96 | 30.50 | 30.56 |
| HotpotQA (31.07) | Adamas (Ours) | 24.05 | **31.65** | 32.18 | 32.89 | 32.30 | 31.45 | 33.06 | **32.56** | 31.41 |
| | Adamas w/o Hadamard | 1.01 | 1.04 | 0.68 | 0.48 | 0.77 | 2.07 | 5.38 | 13.47 | 25.13 |
| | Adamas-1bit | 19.22 | 24.70 | 29.04 | 31.55 | **33.79** | 31.63 | 32.14 | 32.47 | 31.58 |
| | Adamas-3bit | **27.72** | 31.17 | 31.29 | 31.82 | 32.56 | **32.12** | 33.25 | 32.21 | **31.61** |
| | Adamas w/ L2 distance | 18.73 | 28.83 | **32.58** | **32.91** | 31.15 | 30.70 | 32.66 | 31.86 | 31.10 |
| MultifieldQA (41.64) | Adamas (Ours) | 21.53 | 33.38 | 36.60 | 40.67 | 42.49 | **42.93** | 42.22 | 40.77 | 41.83 |
| | Adamas w/o Hadamard | 2.97 | 2.84 | 2.51 | 5.27 | 8.28 | 18.26 | 27.45 | 39.24 | 41.29 |
| | Adamas-1bit | 23.21 | 29.51 | 35.64 | 38.98 | 40.53 | 41.99 | 42.43 | 41.62 | 42.01 |
| | Adamas-3bit | **30.04** | **36.00** | 36.28 | 40.54 | 41.96 | 42.73 | **43.14** | 41.17 | 41.86 |
| | Adamas w/ L2 distance | 23.70 | 34.11 | **39.81** | **41.41** | **43.08** | 41.60 | 41.28 | **41.65** | **42.33** |
| NarrativeQA (21.23) | Adamas (Ours) | 11.98 | 15.13 | **17.81** | **18.52** | 17.36 | 18.59 | 18.74 | 20.14 | 20.22 |
| | Adamas w/o Hadamard | 1.68 | 1.00 | 1.12 | 0.91 | 0.38 | 1.34 | 3.29 | 6.00 | 13.32 |
| | Adamas-1bit | 5.38 | 9.16 | 14.79 | 15.18 | 17.34 | **19.39** | **19.94** | 20.00 | 20.01 |
| | Adamas-3bit | **12.61** | **15.48** | 17.65 | 16.73 | 17.91 | 19.05 | 19.03 | **20.35** | 20.09 |
| | Adamas w/ L2 distance | 8.83 | 14.50 | 16.06 | 18.39 | **18.37** | 19.20 | 18.54 | 19.74 | **20.33** |
| Qasper (28.89) | Adamas (Ours) | 18.83 | 24.52 | 28.33 | 29.38 | **30.68** | 30.93 | 29.94 | 29.26 | 28.65 |
| | Adamas w/o Hadamard | 1.73 | 2.40 | 2.29 | 4.93 | 9.24 | 18.27 | 26.17 | **31.08** | **30.14** |
| | Adamas-1bit | 18.04 | 20.81 | 24.16 | 28.73 | 29.50 | 30.04 | 30.18 | 30.52 | 28.73 |
| | Adamas-3bit | 21.95 | 26.07 | 26.39 | 30.23 | 30.52 | 31.12 | 29.50 | 29.51 | 28.81 |
| | Adamas w/ L2 distance | **22.41** | **27.04** | **28.70** | **31.36** | 30.35 | **31.23** | **30.80** | 29.36 | 28.86 |
| TriviaQA (84.25) | Adamas (Ours) | 56.77 | 75.13 | 78.91 | 82.95 | **84.67** | 83.99 | 83.36 | 83.75 | 83.95 |
| | Adamas w/o Hadamard | 2.63 | 2.08 | 1.21 | 3.52 | 9.73 | 20.14 | 33.15 | 56.62 | 79.13 |
| | Adamas-1bit | 52.26 | 70.62 | 80.08 | 80.88 | 82.59 | 83.90 | **84.11** | **84.22** | 83.63 |
| | Adamas-3bit | **70.99** | **77.74** | **81.40** | 82.95 | 83.99 | 83.79 | 83.46 | 83.55 | 83.68 |
| | Adamas w/ L2 distance | 37.82 | 66.80 | 80.97 | **83.35** | 84.21 | **84.29** | 83.60 | 84.03 | **84.25** |

