# OpenReview forum: "Adamas: Hadamard Sparse Attention for Efficient Long-context Inference"
_ICLR.cc/2026/Conference — Submitted to ICLR 2026_

### Official Review · Reviewer_W5MJ · 2025-10-21

**Soundness:** 3
**Presentation:** 3
**Contribution:** 2
**Rating:** 8
**Confidence:** 4

**Summary:**

This paper introduces "ADAMAS", a lightweight yet highly accurate sparse attention mechanism with training-free. The key innovation is using Hadamard transform, bucketization and 2-bit compression to produce compact representations, and leverages Manhattan-distance estimation for efficient top-k selections. Compared with the existing train-free method, this method can match the accuracy of full attention and achieve up to 1.5× end-to-end speedups.

**Strengths:**

1. Integrating some existing methods into an elegant solution can achieve better results in existing training-free methods.

2. It is a good idea to use Hadamard transform in similarity estimation/sparse selection, make it to achieve better results within efficient calculation.

3. The paper is clearly structured and well-written. The authors provide a thorough explanation of the research motivation, methodology, and experimental results.

**Weaknesses:**

1. In the efficiency analysis, the kernel analysis can be more detailed. Except for full attention, there is no efficiency comparison with other sparse methods.

2. Why it achieves similar results to full attention, in addition to experimental comparison, it would be better if some explanations were given from other aspects, such as case analysis.

**Questions:**

1. I read the original Quest paper, and the PPL metric for PG19 is very close to full attention. Why does your paper show a discrepancy between the PPL and the original paper? Does the Quest method used here use full cache for the first two layers, as in the original paper?

2. Regarding end-to-end decoding latency comparison, can other methods besides full attention be compared?

3. Why use L1 instead of Dot Product to calculate distance? I guess the dot product is faster to compute.

---

> ### Author Response · Authors · 2025-11-20
> **Response to Reviewer W5MJ (W1 & Q2, Q1)**
>
> ## W1 & Q2: More baselines for efficiency comparison
>
> Thank you for your suggestions. We have added SnapKV and L2 as additional baselines and included their LongBench results, which are presented in Figure 3 and Table 1 in the paper. Adamas consistently outperforms all baselines and achieves accuracy on par with full attention, and even surpasses it in certain cases. Specifically, Adamas achieves over a 20% accuracy improvement over Quest, more than a 30% improvement over SnapKV, and accuracy comparable to L2.
>
> Below we show the detailed results:
>
> ### **Accuracy Results**:
>
> Table X: Average accuracy values (%) of all methods evaluated on LongBench, normalized by full-attention results. Adamas achieves over a 20% accuracy improvement over Quest, more than a 30% improvement over SnapKV, and accuracy comparable to L2.
>
> | Methods          | Adamas (Ours) | Adamas w/ L2 | Quest  | SnapKV | StreamingLLM |
> |------------------|---------------|--------------|--------|--------|--------------|
> | Average Accuracy | 98.89         | 98.22        | 77.09  | 66.23  | 54.04        |
>
>
> ### **Evaluation Results**:
>
> Table Y: End-to-end efficiency (s) evaluated on an A800 80G PCIe GPU, encompassing both prefill and 64-token decoding stages. Adamas achieves up to 1.4× speedup with high accuracy.
>
> | Sequence | Methods / Budget | 256   | 512   | 1024  | 2048  | 4096  | Full Attn |
> |----------|-------------------|-------|-------|-------|-------|-------|-----------|
> | **8192** | Adamas (Ours)     | 1.833 | 1.863 | 1.844 | 1.751 | 1.785 | 2.343     |
> |          | Quest             | 1.626 | 1.606 | 1.573 | 1.596 | 1.621 |           |
> |          | SnapKV            | 2.373 | 2.339 | 2.351 | 2.315 | 2.331 |           |
> | **16384**| Adamas (Ours)     | 2.828 | 2.834 | 2.861 | 2.894 | 2.924 | 3.930     |
> |          | Quest             | 2.473 | 2.536 | 2.503 | 2.492 | 2.512 |           |
> |          | SnapKV            | 3.851 | 3.934 | 3.901 | 3.938 | 3.914 |           |
> | **32768**| Adamas (Ours)     | 5.680 | 5.711 | 5.703 | 5.730 | 5.804 | 7.978     |
> |          | Quest             | 4.901 | 4.908 | 4.918 | 4.945 | 4.984 |           |
> |          | SnapKV            | 7.966 | 7.945 | 7.953 | 7.976 | 7.967 |           |
>
> ## Q1:  The discrepancy of the PPL between our paper and Quest paper
>
> Thank you for raising this concern. We clarify that all our experiments were conducted using the same configuration as specified in the Quest paper. Regarding the discrepancy in results: Quest only reported PPL for a token budget of 4096, under which their method achieved performance close to full attention. In our work, we extended the evaluation to more aggressive token budgets (e.g., 256, 512, 1024, and 2048), where Adamas consistently outperforms Quest, demonstrating clear advantages in extremely sparse regimes. For completeness, we have also evaluated Adamas under the 4096-token budget, and the corresponding results are provided in Appendix D. As shown in the Figure 9, Adamas achieves noticeably lower perplexity than full attention, indicating that it can effectively filter out noisy tokens, select the most critical ones, and perform more accurate inference.

---

> ### Author Response · Authors · 2025-11-20
> **Response to Reviewer W5MJ (W2, part of Q3)**
>
> ## W2: Explanation of why Adamas achieves similar results to full attention
>
> Thank you for raising this concern. In the coming days, we will add 3–5 additional case studies to the appendix. We offer the following explanation for why Adamas can achieve performance comparable to, and occasionally surpassing, full attention.
>
> We begin with a simple premise: even if a globally optimal model exists, the trained model is only an approximation of that ideal. Full attention is merely one inference procedure applied to this approximate model. It does not guarantee optimal utilization, especially in extremely long contexts. Therefore, it is possible for a structured sparse mechanism to yield better performance by mitigating the model’s inherent attention biases or inefficiencies.
>
> Prior work, most notably "Lost in the Middle: How Language Models Use Long Contexts" [1], shows that large language models tend to disproportionately attend to the begining and end of the context while under-attending to information in the middle. When the context sequence becomes very long, crucial evidence may be diluted among irrelevant content and effectively “buried,” leading to increased perplexity or incorrect reasoning.
>
> Adamas directly counters this issue by explicitly retrieving high-relevance tokens via its sparse similarity mechanism, preventing such key information from being overwhelmed by long-range noise.
>
> Consider a QA-style task where the key evidence lies in the middle of a 32K-token document, full attention distributes weights across the entire sequence; due to the well-documented positional bias, the model tends to allocate insufficient attention mass to the mid-context evidence, causing the model to either:
>
>   (a) default to generic responses, or
>
>   (b) focus on semantically irrelevant but positionally favored tokens near the beginning or end.
>
> In contrast, Adamas identifies the relevant mid-context tokens via its sparse retrieval step and focuses computation on these segments. In our experiments, we observe that:
>
> - Full attention produces diffuse attention maps with no clear concentration on the supporting evidence.
> - Adamas reliably recalls the correct evidence tokens, enabling the model to produce the accurate answer.
>
> This yields lower perplexity and higher task accuracy, consistent with our empirical results.
> In summary, Adamas’s sparse attention mechanism is not merely an efficiency mechanism, it partially corrects the model’s intrinsic long-context attention bias by resurfacing relevant information that full attention tends to overlook. This explains why Adamas can match, and sometimes exceed, the performance of full attention on long-context reasoning tasks.
>
> ### **Reference**
>
> [1] Nelson F. Liu, Kevin Lin, John Hewitt, Ashwin Paranjape, Michele Bevilacqua, Fabio Petroni, and Percy Liang. 2024. Lost in the Middle: How Language Models Use Long Contexts. Transactions of the Association for Computational Linguistics, 12:157–173.
>
>
> ## Q3: The reason of replacing Dot Product with L1
>
> Thank you for raising this concern. We address it by first explaining the computational motivation and then providing theoretical and empirical justification for using Manhattan (L1) distance as a proxy for dot-product attention.
>
> 1. **Why L1 Distance Is Faster in Adamas**
>
> Although dot products are typically faster on standard hardware, this is not the case under the Adamas pipeline. Adamas compresses queries and keys into 2-bit bucketized codes. To compute dot products using hardware-supported kernels, the 2-bit values would need to be dequantized to 4-bit, since no native 2-bit GEMM kernels exist. Such dequantization introduces substantial overhead and negates the benefits of aggressive compression.
>
> In contrast, we developed a custom high-performance kernel optimized explicitly for computing L1 (Manhattan) distance on 2-bit bucketized codes. This design allows Adamas to:
>
> - avoid dequantization entirely,
> - compute similarity scores directly in compressed space, and
> - achieve significant speedup compared with performing dequantized QK dot products.
>
> Thus, L1 distance is selected primarily for efficiency, being directly compatible with our 2-bit representation and enabling fast token selection with minimal I/O and compute overhead.
>
> 2. **Why L1 Distance Can Replace the Dot Product for Token Selection**
>
> Adamas does not claim that L1 distance is a mathematically exact substitute for the dot product. Sparse attention requires only a relative ranking of tokens with respect to a query. A better order of the keys could lead to selecting the more relevant tokens. Our method leverages negative Manhattan (L1) distance computed on compact 2-bit bucketized codes as an efficient proxy for this ranking.
>
> Below we provide a short mathematical justification showing why this proxy preserves the ordering used by dot-product attention up to controlled perturbations, followed by empirical results validating the approximation quality in practice.

---

> ### Author Response · Authors · 2025-11-20
> **Response to Reviewer W5MJ (part of Q3)**
>
> ## Q3: The reason of replacing Dot Product with L1 (cont'd)
>
> **Mathematical justification (proof sketch)**
>
> (1) Dot product v.s. squared Euclidean distance.
>
> For any query $q$ and key $k$ in $R^d$, the identity holds:
>
>  $$qk^\top = \frac{1}{2}(||q||_2^2 + ||k||_2^2 - ||q-k||_2^2)$$
>
> Hence, for a fixed query $q$, maximizing the dot product $qk^\top$ is equivalent to minimizing the squared Euclidean distance $||q-k||_2^2$, provided that key norms $||k||_2$ does not vary significantly. In modern LLMs with RMSNorm or LayerNorm, key vectors indeed have norms highly concentrated around $\sqrt{d}$, making $||k||_2^2$ a nearly constant, slowly varying term. **Therefore, the ranking is dominated by $||q-k||_2^2$**.
>
> (2) Correlation between $L_1$ and $L_2$ in high dimensions.
>
> Although the two norms satisfy:
>
> $$||v||_2 \le ||v||_1 \le \sqrt{d}||v||_2$$
>
> the key geometric fact in high-dimensional spaces is **strong rank correlation** between $L_1$ and $L_2$:
>
> - Norm equivalence places tight upper/lower bounds on divergence.
> - Statistical proxy effect: unless vectors differ in extreme sparsity patterns (e.g., one-hot vs uniform), $L_1$ and $L_2$ distances concentrate and produce similar **relative orderings**.
> Thus $L_1$ is generally a **robust surrogate** for $L_2$-based rankings.
>
> (3) Combining the pieces (ordering preservation).
>
> Putting (1) and (2) together:
>
> $$max(qk^\top) \Longleftrightarrow min||q−k||_2 ≈ _{proxy} min||q−k||_1.$$
>
> Let $k_a,k_b$ be two keys; if
>
> $$||q-k_a||_1 + \Delta < ||q-k_b||_1$$
>
> and the gap $\Delta>0$ exceeds the bounded distortion from:
>
>   1. the $L_1 \leftrightarrow L_2$ approximation variability
>   2. bucketization perturbation
>
> then the original ordering is perseved:
>
> $$qk_a^\top > qk_b^\top$$
>
> Thus, when pairwise distance gaps exceed the approximation error, the Manhattan proxy produces the same selection top-K selections as dot product.
>
> (4) Effect of 2-bit bucketization.
>
> If bucketization perturbs each coordinate by at most $\varepsilon$, then the induced errors satisfy:
>
> $$\text{$L_1$ error} \le d\varepsilon,\qquad \text{$L_2$ error} \le \sqrt{d}\varepsilon,$$
>
> Thus the ranking errors remain bounded and small provided:
>
> - bucketization keeps $\varepsilon$ small relative to inter-key separation
> - the bucket design preserves large $L_1$ gaps (which dominate attention selection)
>
> Consequently, even coarse 2-bit aggregated codes can still reliably recover the top-ranked keys under dot-product attention. If there are any issues with this analysis, we welcome further discussion. We also hope to refine this aspect as thoroughly as possible.
>
> **Empirical validation**
>
> We corroborate the theoretical argument with experiments: in our paper, Figure 3 shows that (i) Manhattan-based selection on our bucketized 2-bit codes yields token rankings that are consistent with dot-product ranking in the vast majority of cases, and (ii) the downstream accuracy (measured on LongBench) is comparable to or better than Quest under the same sparsity budget. These empirical observations indicate that the bounded approximations above are sufficient in practice for high-quality sparse-attention selection.
>
> **Summary**
>
> We have (a) added the above concise derivation and norm-bound argument to the appendix, (b) included the simple error bounds for the 2-bit bucketization procedure, and (c) referenced Figure 3 as empirical evidence. This combined theoretical + empirical presentation clarifies why negative Manhattan distance on compact, bucketized codes is an effective and efficient proxy for dot-product scoring in Adamas.

---

### Official Review · Reviewer_jitG · 2025-10-29

**Soundness:** 2
**Presentation:** 2
**Contribution:** 2
**Rating:** 2
**Confidence:** 4

**Summary:**

The paper proposes a sparse attention mechanism for long-context LLM inference that combines Hadamard transforms, 2-bit bucketization, and Manhattan distance estimation to select relevant key-value pairs. The authors claim up to 8× higher sparsity than prior methods while maintaining accuracy comparable to full attention. The paper reads more like an application of previously known methods to sparse attention rather than a fundamental innovation.

**Strengths:**

- The method shows consistent improvements over baselines across multiple benchmarks  with good speedups.
- The paper includes perplexity, accuracy, and efficiency metrics, plus ablation studies validating each component.
- Custom CUDA kernels demonstrate real-world feasibility with actual latency measurements.

**Weaknesses:**

- ( Contribution) The core idea of using Hadamard transforms to smooth distributions before quantization is borrowed directly from QuaRot. The novelty claim is therefore quite weak.
- (Method) "Bucketization" looks a lot like standard quantization.
- (Baselines) The comparison is limited to StreamingLLM, that is a basic sliding window method that performs poorly on retrieval tasks by design and Quest, that performs page-level selection with known coarse-grained limitations. A lot of Sparse Attention and KV Cache compression methods have been ignored. For example SnapKV, L2 Norm, MInference (mentioned but not compared) and DUOAttention (mentioned but not benchmarked).
- In Equation 6 why does negative Manhattan distance on 2-bit quantized codes approximate dot products in the original space? The paper provides no theoretical or empirical justification for this.
- There is no explanation concerning the choice for the bucketization thresholds. Also, there is no analysis about how sensitive is performance to different thresholds.

**Questions:**

- In fig.4 it looks like perplexity with Adamas is lower than full attention, how do the authors explain this ?
- Can you provide theoretical justification for why Manhattan distance on quantized codes approximates attention scores?
- How were bucketization thresholds chosen?
- What is the difference between sparse attention and kv cache compression methods ? How do they affect differently memory and latency ? The paper would benefit from this clarification.

---

> ### Author Response · Authors · 2025-11-20
> **Response to Reviewer jitG (W1 & W2)**
>
> ## W1 & W2: Contribution and details of Adamas
>
> Thank you for raising these concerns. We first demonstrate that the novelty of Adamas is clear by explaining its design. Adamas is a sparse attention method that achieves nearly lossless performance while using only 128 selected tokens during the decoding-stage attention computation. Under this setting, Adamas delivers up to $4.4 \times$ speedup for self-attention and $1.5\times$ end-to-end acceleration on 32K-length sequences. As illustrated in Figure 2, Adamas is not based on a single technique. Rather, it integrates multiple components, including the Hadamard transform, bucketization, 2-bit compression, and Manhattan distance-based similarity estimation, to form an effective and efficient complete sparse attention pipeline.
>
> The Hadamard transform itself is a classical linear transform based on orthogonal Hadamard matrices, originally introduced by Jacques Hadamard in 1893 [1]. It has been widely used across various fields such as signal processing, data compression, and quantum computing, etc. While QuaRot employs the Hadamard transform primarily to reduce outlier effects for better quantization, our use case is different. We apply the transform to uniformly disperse query and key information across dimensions, which improves the robustness and accuracy of similarity estimation. Thus, the use of the Hadamard transform in Adamas is neither introduced nor conceptually dependent on QuaRot.
>
> Regarding bucketization, we emphasize that our bucketization mechanism is fundamentally different from standard quantization. Quantization methods such as FP8 typically apply blockwise or groupwise quantization, mapping values into FP8 while maintaining high-precision scaling factors. These scaling factors allow the quantized values to be efficiently dequantized back to high-precision formats such as BF16, enabling hardware-accelerated low-bit GEMM kernels to achieve both efficiency and accuracy.
>
> In contrast, bucketization does not perform quantization at all. Instead of approximating numerical values using a low-bit floating-point format, bucketization groups the query/key vectors according to their magnitudes and assigns each token to a group using only a 2-bit group index. Crucially, unlike FP8-style quantization:
> - Bucketization has no scaling factors.
> - Bucketized tokens cannot be restored to the original BF16/FP16 precision.
> - The 2-bit codes encode only coarse group membership, not numerical value.
>
> Thus, bucketization serves a completely different purpose: it provides a highly compact structural representation used solely for estimating Manhattan distances during token selection, not for reconstructing high-precision vectors or accelerating GEMM.
>
> Because Adamas is designed as a sparse attention method whose goal is to preserve LLM performance while operating on only a small set of selected tokens. Prior studies on static sparse patterns [2-5] show that the quality of the KV-token selection algorithm is critical for maintaining the performance. However, any sparse attention method must evaluate selection criteria over the entire sequence for selection, which incurs $O(T)$ complexity and is prohibitively expensive when operating directly in BF16/FP16. To address this issue, Adamas introduces bucketization together with 2-bit compression to drastically reduce I/O overhead and significantly accelerate the Manhattan distance estimation (used for token selection). Moreover, because 2-bit arithmetic lacks native hardware support, we further implemented a custom high-performance kernel (please see Figure 5 in our paper) to make the pipeline efficient in practice. Overall, both the objectives and the underlying mechanisms of Adamas differ fundamentally from standard quantization approaches.
>
> ### **References**
>
> [1] Hadamard, Jacques. "Résolution d'une question relative aux déterminants." Bull. des sciences math. 2 (1893): 240-246.
>
> [2] Zhang, Zhenyu, et al. "H2o: Heavy-hitter oracle for efficient generative inference of large language models." Advances in Neural Information Processing Systems 36 (2023): 34661-34710.
>
> [3] Xiao, Chaojun, et al. "Infllm: Training-free long-context extrapolation for llms with an efficient context memory." Advances in Neural Information Processing Systems 37 (2024): 119638-119661.
>
> [4] Xiao, Guangxuan, et al. "Efficient streaming language models with attention sinks." arXiv preprint arXiv:2309.17453 (2023).
>
> [5] Xiao, Guangxuan, et al. "Duoattention: Efficient long-context llm inference with retrieval and streaming heads." arXiv preprint arXiv:2410.10819 (2024).

---

> ### Author Response · Authors · 2025-11-20
> **Response to Reviewer jitG (W3)**
>
> ## W3: More baselines for comparison
>
> Thank you for your suggestions. We have added SnapKV and L2 as additional baselines and included their LongBench results in Figure 3 and Table 1 in the paper. Since MInference focuses on accelerating the prefill stage while Adamas targets the decoding stage, the two methods are orthogonal and can, in principle, be combined. Therefore, we do not consider MInference a directly comparable baseline. For DUOAttention, we cannot find similar results in their paper. It also requires additional training phase, whereas Adamas and other baselines are in category of training-free method, thus we do not include DUOAttention as a baseline. The evaluation results are presented in Figure 3 and Table 1 in the paper. Adamas consistently outperforms all baselines and achieves accuracy on par with full attention, and even surpasses it in certain cases. Specifically, Adamas achieves over a 20% accuracy improvement over Quest, more than a 30% improvement over SnapKV, and accuracy comparable to L2.
>
> Detailed results are shown below:
>
> ### **Accuracy Results**:
>
> Table X: Average accuracy values (%) of all methods evaluated on LongBench, normalized by full-attention results. Adamas achieves over a 20% accuracy improvement over Quest, more than a 30% improvement over SnapKV, and accuracy comparable to L2.
>
> | Methods          | Adamas (Ours) | Adamas w/ L2 | Quest  | SnapKV | StreamingLLM |
> |------------------|---------------|--------------|--------|--------|--------------|
> | Average Accuracy | 98.89         | 98.22        | 77.09  | 66.23  | 54.04        |
>
>
> ### **Evaluation Results**:
>
> Table Y: End-to-end efficiency (s) evaluated on an A800 80G PCIe GPU, encompassing both prefill and 64-token decoding stages. Adamas achieves up to 1.4× speedup with high accuracy.
>
> | Sequence | Methods / Budget | 256   | 512   | 1024  | 2048  | 4096  | Full Attn |
> |----------|-------------------|-------|-------|-------|-------|-------|-----------|
> | **8192** | Adamas (Ours)     | 1.833 | 1.863 | 1.844 | 1.751 | 1.785 | 2.343     |
> |          | Quest             | 1.626 | 1.606 | 1.573 | 1.596 | 1.621 |           |
> |          | SnapKV            | 2.373 | 2.339 | 2.351 | 2.315 | 2.331 |           |
> | **16384**| Adamas (Ours)     | 2.828 | 2.834 | 2.861 | 2.894 | 2.924 | 3.930     |
> |          | Quest             | 2.473 | 2.536 | 2.503 | 2.492 | 2.512 |           |
> |          | SnapKV            | 3.851 | 3.934 | 3.901 | 3.938 | 3.914 |           |
> | **32768**| Adamas (Ours)     | 5.680 | 5.711 | 5.703 | 5.730 | 5.804 | 7.978     |
> |          | Quest             | 4.901 | 4.908 | 4.918 | 4.945 | 4.984 |           |
> |          | SnapKV            | 7.966 | 7.945 | 7.953 | 7.976 | 7.967 |           |

---

> ### Author Response · Authors · 2025-11-20
> **Response to Reviewer jitG (W4 & Q2)**
>
> ## W4 & Q2: Theoretical or empirical justification
>
> Thank you for raising this concern. Below we post the justification from both theoretical and empirical aspects in terms of Manhattan distance selected in Adamas.
>
> First of all, Adamas does not claim that negative Manhattan distance is a mathematically exact replacement for the dot product. Instead, sparse attention only requires a relative ranking (a partial order) of keys with respect to a query. A better order of the keys could lead to selecting the more relevant tokens. Our method leverages negative Manhattan (L1) distance computed on compact 2-bit bucketized codes as an efficient proxy for this ranking. Below we provide a short mathematical justification showing why this proxy preserves the ordering used by dot-product attention up to controlled perturbations, followed by empirical results (Figure 3) validating the approximation quality in practice.
>
> **Mathematical justification (proof sketch)**
>
> (1) Dot product v.s. squared Euclidean distance.
>
> For any query $q$ and key $k$ in $R^d$, the identity holds:
>
>  $$qk^\top = \frac{1}{2}(||q||_2^2 + ||k||_2^2 - ||q-k||_2^2)$$
>
> Hence, for a fixed query $q$, maximizing the dot product $qk^\top$ is equivalent to minimizing the squared Euclidean distance $||q-k||_2^2$, provided that key norms $||k||_2$ does not vary significantly. In modern LLMs with RMSNorm or LayerNorm, key vectors indeed have norms highly concentrated around $\sqrt{d}$, making $||k||_2^2$ a nearly constant, slowly varying term. **Therefore, the ranking is dominated by $||q-k||_2^2$**.
>
> (2) Correlation between $L_1$ and $L_2$ in high dimensions.
>
> Although the two norms satisfy:
>
> $$||v||_2 \le ||v||_1 \le \sqrt{d}||v||_2$$
>
> the key geometric fact in high-dimensional spaces is **strong rank correlation** between $L_1$ and $L_2$:
>
> - Norm equivalence places tight upper/lower bounds on divergence.
> - Statistical proxy effect: unless vectors differ in extreme sparsity patterns (e.g., one-hot vs uniform), $L_1$ and $L_2$ distances concentrate and produce similar **relative orderings**.
> Thus $L_1$ is generally a **robust surrogate** for $L_2$-based rankings.
>
> (3) Combining the pieces (ordering preservation).
>
> Putting (1) and (2) together:
>
> $$max(qk^\top) \Longleftrightarrow min||q−k||_2 ≈ _{proxy} min||q−k||_1.$$
>
> Let $k_a,k_b$ be two keys; if
>
> $$||q-k_a||_1 + \Delta < ||q-k_b||_1$$
>
> and the gap $\Delta>0$ exceeds the bounded distortion from:
>
>   1. the $L_1 \leftrightarrow L_2$ approximation variability
>   2. bucketization perturbation
>
> then the original ordering is perseved:
>
> $$qk_a^\top > qk_b^\top$$
>
> Thus, when pairwise distance gaps exceed the approximation error, the Manhattan proxy produces the same selection top-K selections as dot product.
>
> (4) Effect of 2-bit bucketization.
>
> If bucketization perturbs each coordinate by at most $\varepsilon$, then the induced errors satisfy:
>
> $$\text{$L_1$ error} \le d\varepsilon,\qquad \text{$L_2$ error} \le \sqrt{d}\varepsilon,$$
>
> Thus the ranking errors remain bounded and small provided:
>
> - bucketization keeps $\varepsilon$ small relative to inter-key separation
> - the bucket design preserves large $L_1$ gaps (which dominate attention selection)
>
> Consequently, even coarse 2-bit aggregated codes can still reliably recover the top-ranked keys under dot-product attention. If there are any issues with this analysis, we welcome further discussion. We also hope to refine this aspect as thoroughly as possible.
>
> **Empirical validation**
>
> We corroborate the theoretical argument with experiments: in our paper, Figure 3 shows that (i) Manhattan-based selection on our bucketized 2-bit codes yields token rankings that are consistent with dot-product ranking in the vast majority of cases, and (ii) the downstream accuracy (measured on LongBench) is comparable to or better than Quest under the same sparsity budget. These empirical observations indicate that the bounded approximations above are sufficient in practice for high-quality sparse-attention selection.
>
> **Summary**
>
> We have (a) added the above concise derivation and norm-bound argument to the appendix, (b) included the simple error bounds for the 2-bit bucketization procedure, and (c) referenced Figure 3 as empirical evidence. This combined theoretical + empirical presentation clarifies why negative Manhattan distance on compact, bucketized codes is an effective and efficient proxy for dot-product scoring in Adamas.

---

> ### Author Response · Authors · 2025-11-20
> **Response to Reviewer jitG (W5 & Q3)**
>
> ## W5 & Q3: Analysis of bucketization threshold selection
>
> Thank you for your question. Below we provide a detailed explaination of the motivation and empirical analysis behind our choice of bucketization thresholds.
>
> 1. **Distributional motivation for choosing bucketization thresholds.**
>
> We examined the distribution of RoPE-transformed Q/K vectors on LongBench and found that each dimension approximately follows a zero-mean normal distribution (detailed plots and statistics are included in the Appendix C). Based on this observation, we evaluated different bucketization thresholds (b) under a representative setting of token_budget = 128, where the bucketization is defined by the rule $[-b, 0, b]$. The results are shown in Table Z. Notice that this motivation is based on the observation from LongBench, the performance of Adamas still generalize to other benchmark such as NIAH, where the new added experiments supporting this.
>
> We shown the detailed results below:
>
> Table Z: Sensitivity study of the bucketization threshold, evaluated on six LongBench datasets. Top-2 scores are bolded. Thresholds of 1 and 10 show consistently better accuracy.
>
> | Dataset / Threshold | 1        | 3       | 5       | 7        | 10       | 15       |
> |---------------------|----------|---------|---------|----------|----------|----------|
> | GovReport           | **31.06** | 25.29  | 29.99  | 29.62   | **30.44** | 29.20    |
> | HotpotQA            | **33.38** | 26.12  | 30.81  | 29.00   | **32.89** | 31.54    |
> | MultifieldQA        | 40.28    | 33.54  | 37.75  | **41.65** | **40.67** | 39.35    |
> | NarrativeQA         | **17.96** | 16.47  | 15.45  | 16.02   | **18.52** | 14.90    |
> | Qasper              | **28.95** | 23.47  | 27.15  | 28.16   | **29.38** | **29.38** |
> | TriviaQA            | **83.39** | 74.70  | 80.46  | 81.04   | **82.95** | 80.65    |
>
> 2. **Empirical sensitivity results.**
>
> Our experiments show that:
>   - **$b = 1$** (fine-grained threshold in implicit standardized units by LayerNorm), and
>   - **$b = 10$** (coarse-grained threshold in raw magnitude)
>
> both achieve strong performance.
>
> Thresholds in the mid-range (e.g., $b = 3, 5, 7$) or higher than 10 show moderate degradation but still remain functional.
>
> 3. **Comprehensive LongBench Comparison for $b = 1$ and $b = 10$.**
>
> To further validate the effectiveness of the two best-performing thresholds, we conduct comprehensive evaluations of $b=1$ and $b=10$ across the entire LongBench benchmark. The results, presented in Figure 9, show that the performance curves of $b=1$ and $b=10$ almost completely overlap on every task. Moreover, both thresholds significantly outperform all other baselines, demonstrating that these two bucketization schemes consistently preserve the essential features required for accurate QK ranking under sparse attention.
>
> 4. **Why both $b = 1$ and $b = 10$ work well.**
>
> These two choices correspond to two meaningful ways of capturing the dominant contributors to the QK dot product in complementary ways:
>
>   - **Large-magnitude components** $(|x| > 10)$ contribute disproportionately to the QK dot product and often encode strong similarity signals.
>   - **Near-zero components** $(|x| < 1)$ provide little useful information and behave mostly as noise.
>
> Thus:
>
>   - **$b = 1$** cleanly separates “strong positive / strong negative / near-zero” signals in standardized units, effectively separating strong from weak signals.
>   - **$b = 10$** highlights “large-magnitude features” directly in raw units, capturing the dimensions that dominate the dot product.
>
> Both thresholds preserve the essential structural information needed to approximate QK ranking.
>
> 5. **Why intermediate or larger thresholds perform worse.**
>
> Mid-range thresholds may:
>
>   - over-partition weak-signal regions (adding noise), or
>   - over-merge strong-signal regions (losing contrast),
>
> leading to slight degradation in similarity estimation.
>
> 6. **Conclusion.**
>
> Given the near-normal distribution of Q/K dimensions, the bucketization threshold only needs to reliably separate strong-contribution dimensions from weak ones for the ranking to remain robust. Thresholds that fail to achieve this separation exhibit moderate performance loss, whereas $b=1$ and $b=10$ both provide meaningful, interpretable cut points and thus perform well.

---

> ### Author Response · Authors · 2025-11-20
> **Response to Reviewer jitG (Q1, Q4)**
>
> ## Q1: Explanation of "why perplexity with Adamas is lower than full attention in Figure 4"
>
> Thank you for raising this question. Based on the findings from “Lost in the Middle: How Language Models Use Long Contexts” [1], language models tend to rely more on information at the beginning and end of the context, while struggling to effectively utilize information in the middle. As a result, when the sequence length becomes excessively long, key information essential for reasoning may be diluted and obscured by large amounts of irrelevant tokens, leading to higher perplexity under full attention. Adamas employs sparse attention to selectively retrieve the most critical tokens, effectively guiding the model to focus on essential content. By filtering out distracting or low-relevance tokens, Adamas mitigates the "information dilution" effect and can achieve low perplexity than full attention in certain long-context settings.
>
> ### **Reference**
>
> [1] Nelson F. Liu, Kevin Lin, John Hewitt, Ashwin Paranjape, Michele Bevilacqua, Fabio Petroni, and Percy Liang. 2024. Lost in the Middle: How Language Models Use Long Contexts. Transactions of the Association for Computational Linguistics, 12:157–173.
>
> ## Q4: The difference between sparse attention and kv cache compression methods
>
> Thank you for raising this question. Below we summarize the key differences between sparse attention and KV cache compression, and the summary has been added in the revised version.
>
> 1. **Baseline complexity.** The time and space complexity of decoding with KV cache is typically $O(T)$, where $T$ represents sequence length.
>
> 2. **Sparse attention (focus: latency reduction).** Sparse attention focuses on selecting the most relevant query-key pairs at the attention level, with the overall goal of approximating full attention computation using a subset of tokens to achieve acceleration. This reduces I/O during attention computation, especially under high sparsity, and thus primarily improves latency (i.e., reduces time complexity), rather than memory usage.
>
> 3. **KV cache compression (focus: memory reduction).** KV cache compression reduces the memory footprint of stored key/value tensors (space complexity). However, because compression or pruning of tokens is generally irreversible, such methods often incur accuracy loss and may introduce additional decoding latency. Moreover, simply compressing the KV cache alone may not necessarily reduce I/O during attention computation, limiting its effectiveness for speedup.
>
> 4. **Overall relationship.** These two approaches are complementary: sparse attention primarily targets computational latency, while KV cache compression focuses on memory reduction. Each addresses a different bottleneck in efficient long-context decoding.

---

### Official Review · Reviewer_M26R · 2025-10-31

**Soundness:** 3
**Presentation:** 3
**Contribution:** 2
**Rating:** 6
**Confidence:** 4

**Summary:**

This paper proposes Adamas, a training‑free, token‑level sparse attention mechanism for long‑context LLM inference. The core idea is to apply an orthogonal Hadamard transform to queries and keys, then bucketize the transformed vectors into 2‑bit codes that are stored alongside the KV cache. During decoding, Adamas uses a lightweight Manhattan‑distance estimator on these 2‑bit codes to select top‑k candidates, followed by sparse attention on the reduced set (Figure 2, p.3; Algorithm 1, p.4). The method aims to preserve accuracy while enabling higher sparsity and end‑to‑end speedups. Empirically, the paper claims parity with full attention at small token budgets (near‑lossless at 128) and reports up to 4.4× self‑attention speedup and 1.5× end‑to‑end speedup on 32K sequences, sometimes with perplexity even lower than full attention.

**Strengths:**

- The pipeline couples an orthogonal transform that smooths activations with low‑bit quantization, enabling efficient integer-only L1 screening before exact attention. The identity \((QH)(KH)^\top = QK^\top\) justifies operating in the Hadamard basis without loss.

- Token‑level dynamic selection: Unlike Quest’s page‑level granularity, Adamas selects at the token level, which plausibly improves recall under tight budgets. The LongBench curves show the smallest gap to full attention at low budgets across GovReport, HotpotQA, MultifieldQA, NarrativeQA, Qasper, TriviaQA.

- Custom CUDA kernels for fused bucketization+compression, Manhattan-distance estimation, top‑k, and sparse attention are provided; kernel breakdowns demonstrate speedups vs. FlashInfer baselines.

**Weaknesses:**

- The paper does not explicitly describe how Adamas operates during the prefill stage or whether the proposed sparse attention strategy is applied there. If the sparse selection is only used during decoding while the prefill phase still relies on full dense attention, then evaluations on datasets like LongBench—whose inputs involve long prefill sequences—may not fully reflect the accuracy implications of the sparse mechanism. Clarifying whether Adamas affects both prefill and decoding phases (or only the latter) is essential for interpreting the reported efficiency and accuracy trade-offs.

- While LongBench, PG19, and passkey retrieval are helpful, the paper omits widely used Needle‑in‑a‑Haystack stress tests that specifically probe long‑range retrieval under adversarial distractors. Adding these would directly support the “better recall under high sparsity” claim.

- Dynamic/adaptive inference baselines are limited mainly to Quest; other competitive methods targeting inference‑time sparsity appear in related work but are not compared head‑to‑head in decoding regimes.

- The efficiency study is on an RTX A6000. Results on A100/H100 would be informative for server‑side deployment and for understanding tensor‑core utilization, especially given the bit‑wise kernels.

- The figure annotates PPL/Acc values but the exact setup is not fully specified in‑figure. A precise caption or footnote would aid interpretability.

**Questions:**

- Figure 1 and complexity: Could you specify the dataset(s) and token budgets behind the PPL/Acc annotations? And in my opinion, both Adamas and Quest are \(O(L)\) like vanilla attention.

- The most critical clarification is whether the proposed sparse attention mechanism is applied during both the prefill and decoding stages, or only during decoding. If Adamas is designed specifically for decoding and the reported speedups are measured only in that phase, while the benchmark datasets (e.g., LongBench) are dominated by prefill-heavy workloads, the results may not convincingly demonstrate end-to-end acceleration.

- Additional detailed questions can be found in the Weaknesses section.

---

> ### Author Response · Authors · 2025-11-20
> **Response to Reviewer M26R (W1 & Q2)**
>
> Thank you for the thoughtful comments. Below we provide our detailed responses to each concern. For closely related questions and concerns, we provide a combined clarification.
>
> ## W1 & Q2: The scope of Adamas and the effectiveness of selecting LongBench
> Thank you for raising this point. We clarify that Adamas, along with other baseline methods (training-free sparse attention), requires a prefill stage to compute the full KV cache first. During decoding, Adamas selects the most important tokens among all the input sequences and retrieves their KV cache for sparse attention computation, thereby achieving sparsity and acceleration. Notice that this process is standard for this category of sparse attention methods. Following the practice of Quest, we employ full attention during the prefill stage and apply our proposed Adamas algorithm during the decoding stage.
>
> Our goal is to accelerate the decoding phase as much as possible under the constraint of maintaining performance. Although LongBench indeed contains very long prefill inputs, the decoding phase remains equally critical for evaluating sparse-attention methods.  In LongBench benchmark, after the prefill stage computes the full KV cache, Adamas must, under a limited token budget (e.g., 256 tokens), repeatedly and accurately identify the most relevant keys from the entire sequence that average 5K-15K tokens in length. Even the output sequences are only a few dozen tokens, this high-precision retrieval, recovering 256 tokens from several thousand, must be executed tens of times to generate a high-quality answer. Adamas is designed to address this problem. In our evaluation on LongBench, Adamas successfully outperforms all baselines and matches the performance of full attention. Therefore, we believe that the LongBench provides a meaningful and stringent assessment of decoding-stage accuracy.
>
> Regarding end-to-end generation efficiency, we have updated our evaluation results in Appendix B (temporarily posted in Appendix of our revised version, will be moved to the main page later) to include both the prefill and decoding stages, directly reflecting the acceleration achieved in practical deployment behavior. It is worth noting that Quest adopts page-wise sparsity and tends to sacrifice accuracy for speed, especially under high-sparsity regimes, whereas Adamas consistently delivers substantially high accuracy, over 20% improvement against Quest (please see Table 1 in the paper). Although operating at the fine-grained token-wise level, Adamas is only about 10% slower than Quest, underscoring the effectiveness of our carefully optimized high-performance kernels.
>
> Detailed results are shown below:
>
> ### **Accuracy Results**:
>
> Table X: Average accuracy values (%) of all methods evaluated on LongBench, normalized by full-attention results. Adamas achieves over a 20% accuracy improvement over Quest, more than a 30% improvement over SnapKV, and accuracy comparable to L2.
>
> | Methods          | Adamas (Ours) | Adamas w/ L2 | Quest  | SnapKV | StreamingLLM |
> |------------------|---------------|--------------|--------|--------|--------------|
> | Average Accuracy | 98.89         | 98.22        | 77.09  | 66.23  | 54.04        |
>
>
> ### **Evaluation Results**:
>
> Table Y: End-to-end efficiency (s) evaluated on an A800 80G PCIe GPU, encompassing both prefill and 64-token decoding stages. Adamas achieves up to 1.4× speedup with high accuracy.
>
> | Sequence | Methods / Budget | 256   | 512   | 1024  | 2048  | 4096  | Full Attn |
> |----------|-------------------|-------|-------|-------|-------|-------|-----------|
> | **8192** | Adamas (Ours)     | 1.833 | 1.863 | 1.844 | 1.751 | 1.785 | 2.343     |
> |          | Quest             | 1.626 | 1.606 | 1.573 | 1.596 | 1.621 |           |
> |          | SnapKV            | 2.373 | 2.339 | 2.351 | 2.315 | 2.331 |           |
> | **16384**| Adamas (Ours)     | 2.828 | 2.834 | 2.861 | 2.894 | 2.924 | 3.930     |
> |          | Quest             | 2.473 | 2.536 | 2.503 | 2.492 | 2.512 |           |
> |          | SnapKV            | 3.851 | 3.934 | 3.901 | 3.938 | 3.914 |           |
> | **32768**| Adamas (Ours)     | 5.680 | 5.711 | 5.703 | 5.730 | 5.804 | 7.978     |
> |          | Quest             | 4.901 | 4.908 | 4.918 | 4.945 | 4.984 |           |
> |          | SnapKV            | 7.966 | 7.945 | 7.953 | 7.976 | 7.967 |           |

---

> ### Author Response · Authors · 2025-11-20
> **Response to Reviewer M26R (W2, W3, W4, W5 & Q1)**
>
> ## W2: More experiments on Needle-in-a-Haystack stress tests
>
> Thank you for your advice. We have evaluated Adamas and the baselines on the Needle-in-a-Haystack stress tests. Using the Yarn-Llama-2-7b-128k model at a context length of 32K, we compared the recall rates (measured by ROUGE scores) of Adamas, Quest, and full attention . As shown in Appendix E (temporarily posted in Appendix of our revised version, will be moved to the main page later), Adamas achieves recall performance nearly identical to that of full attention even under high sparsity with only 256-token budget, consistently outperforming the baseline. Specifically, both Adamas and full attention achieve scores around 7/10 at a sequence length of 32K, whereas Quest reaches only slightly above 2. These results provide strong and direct evidence supporting our claim of "better recall under high sparsity".
>
> ## W3: More evaluation compared to other competitive methods
>
> Thank you for the question. We have additionally included SnapKV and L2 as comparative baselines. As shown in Figure 3, Adamas consistently outperforms all baselines and achieves accuracy comparable to full attention, and in some cases even surpasses it. According to Table 1 in the paper, Adamas surpasses Quest by more than 20% on average accuracy values normalized by full attention results, highlighting its clear advantage in maintaining high-quality performance under sparse attention.
>
> Please refer to Table X in "Response to Reviewer M26R (W1 & Q2)" for detailed results.
>
> ## W4: Efficiency study on server‑side deployment
>
> Thank you for the suggestions. To evaluate the efficiency of Adamas on server-side deployment, we rented an A800 80G PCIe GPU to conduct end-to-end efficiency evaluations that better reflect real-world server environments. These results are temporarily included in appendix B. It is worth noting that Quest adopts page-wise sparsity and tends to sacrifice accuracy for speed, especially under high-sparsity regimes, whereas Adamas consistently delivers substantially high accuracy, over 20% improvement against Quest (please see Table 1 in the paper). Although operating at the fine-grained token-wise level, Adamas is only about 10% slower than Quest, underscoring the effectiveness of our carefully optimized high-performance kernels.
>
> Please refer to Table Y in "Response to Reviewer M26R (W1 & Q2)" for detailed results.
>
> ## W5 & Q1: Interpretability enhancement for Figure 1
>
> Thank you for the suggestions. The PPL values annotated in Figure 1 are derived from our experiments on PG19, specifically corresponding to the token budget of 256 reported in Figure 4. The accuracy (Acc) values represent the average performance over the six LongBench subsets, normalized by full attention results (please see Table 1 in the paper), reflecting the overall trends in Figure 3. We appreciate the reviewer's suggestion again and have updated the figure caption in our reviesed version accordingly to clearly indicate the sources of these data in recent days.
> Regarding computational complexity: Although both Adamas and Quest have an overall time complexity of $O(L)$ which is similar to vanilla attention, the key difference under sparse attention lies in how the effective number of tokens participating in computation is reduced. In Adamas, the selected token count $L$ is a small, predefined constant, whereas the full sequence length $T$ continues to grow during decoding phase. Thus $L \ll T$ naturally holds throughout generation. This reduction in effective context length enables the observed decoding-time acceleration, turning attention computation part from linear complexity in $O(T)$ to the constant-cost operation $O(L)$.

---

### Author Response · Authors · 2025-11-27
**Response to all reviewers**

We sincerely appreciate the valuable suggestions provided by all reviewers. We have revised the paper accordingly and have submitted an updated version. In the paper, we have added the following:

1. Additional baselines (SnapKV and L2):

  - Figure 3 and Appendix A now include the accuracy results of SnapKV and L2 on LongBench.
  - Appendix B further reports end-to-end efficiency comparisons among Adamas, Quest, and SnapKV on the A800 GPU.

2. A detailed bucketization threshold analysis in Appendix C, including:

  - empirical distributions of Queries and Keys,
  - accuracy sensitivity under different thresholds, and
  - an analysis explaining the choice of effective threshold values.

3. Needle-In-A-Haystack experimental results in Appendix E, newly added as an additional robustness evaluation.
4. Perplexity curves under a 4096-token budget in Appendix D, demonstrating Adamas’s ability to filter noise tokens and preserve reasoning accuracy.

Regarding SnapKV, we would like to clarify two important issues:

1. Perplexity evaluation is incompatible with SnapKV:
 Perplexity experiments require computing token-level PPL step-by-step. However, [SnapKV’s mechanism](https://github.com/FasterDecoding/SnapKV/blob/e216ddc84c5bd210378cbdbbba12ba02102aa640/snapkv/monkeypatch/snapkv_utils.py#L42) cannot operate correctly under this setting, making PPL evaluation impossible.
2. SnapKV’s official implementation fails on most Passkey test cases:
 Due to issues in SnapKV’s implementation, despite our best efforts to follow the [official examples](https://github.com/FasterDecoding/SnapKV/blob/main/notebooks/example.ipynb) and the implementation used in their [LongBench experiments](https://github.com/FasterDecoding/SnapKV/blob/main/experiments/LongBench/pred_snap.py), SnapKV consistently fails on nearly all Passkey cases. When failure occurs, it silently falls back to FlashAttention, producing results that do not reflect SnapKV’s intended behavior. As such, we cannot reliably include SnapKV in these comparisons.

We thank all reviewers for the insightful comments. If our responses have sufficiently addressed the concerns and provided additional clarity, we would greatly appreciate the reviewers' consideration in raising the score.
Please feel free to let us know if you have any further questions or suggestions. Thank you very much!

---

### Author Response · Authors · 2025-11-30
**Rebuttal Summary and Clarifications for AC Review**

Dear PCs, SACs, and ACs, and Reviewers,

We would like to sincerely thank the PCs, SACs, ACs, and Reviewers for your careful evaluation of our submission.

This note provides a concise, author-side summary of our rebuttal. Our paper addresses the challenge of achieving accurate and efficient sparse attention for long-context LLM inference, proposing Adamas, a training-free method that delivers near-lossless accuracy together with substantial end-to-end acceleration.

In response to the discussion, we have added several supplementary appendices and revised parts of the paper for completeness and thoroughly address the Reviewers' concerns raised during the review process.

We believe our rebuttal thoroughly addresses the Reviewers' concerns and would lead them to form a more positive assessment of our paper. However, due to the data-leak issue on openreview.net, ACs are asked to estimate how the reviewer's impressions would have changed, incursing additional work. To assist with this process, we provide this concise summary to help the AC efficiently and accurately assess the paper and the reviewers’ feedback during the decision-making period.

For the AC’s convenience, the following parts summarize (1) the key strengths highlighted by the Reviewers and (2) how our rebuttal addresses their concerns. Detailed responses can be found in the corresponding sections of the discussion.

We hope this summary assists the AC in efficiently assessing the paper and the reviewers’ feedback. Thank you again for your time and consideration.

Best regards,

The Authors of Submission 2783.

---

> ### Author Response · Authors · 2025-11-30
> **Strengths**
>
> The strengths of our paper are highlighted by the reviewers:
>
> - **Consistent improvements over baselines** with strong speedups and better accuracy (M26R, jitG, W5MJ).
> - **An elegant and well-engineered sparse attention pipeline**, integrating Hadamard transforms, bucketization, 2-bit compression and efficient Manhatton distance-based token-level dynamic selection (M26R, W5MJ).
> - **Custom CUDA kernels** demonstrate **real-world feasibility** with actual latency measurements (M26R, jitG).
> - **Inclusion of** perplexity/accuracy/efficiency metrics and **ablation studies**, validating each component (jitG).
> - **Clear structure and writing quality of the paper**, providing a thorough explanation of the research motivation, methodology, and experimental results (W5MJ).

---

> ### Author Response · Authors · 2025-11-30
> **Summary for the response to weaknesses and questions of Reviewer M26R**
>
> - **W1 & Q2: Prefill vs. Decoding Clarification.** We clarified that, consistent with all training-free sparse attention baselines, Adamas uses dense attention only in prefill and applies sparsity strictly in decoding stage. We further explained why decoding is the key phase for evaluating sparse-attention accuracy on LongBench, added full end-to-end A800 results in Appendix B, and demonstrated Adamas’s strong decoding-stage accuracy and speed.
>
> - **W2: Missing Needle-in-a-Haystack experiments.** We added full Needle-in-a-Haystack (NIAH) experiments (Appendix E), showing that Adamas achieves recall nearly identical to full attention with only a 256-token budget, and significantly outperforms Quest under the same setting.
>
> - **W3: Limited comparison to other dynamic/adaptive sparse-attention baselines.** We added SnapKV and L2 as additional baselines. The updated experimental results (Figure 3 and Table 1 in the revised submission) show that Adamas consistently outperforms both methods and matches or even exceeds full attention on LongBench.
>
> - **W4: Efficiency results only on RTX A6000.** We conducted new end-to-end experiments on an A800 PCIe GPU, demonstrating up to 1.4× speedup with high accuracy, and reported the full results in Appendix B.
>
> - **W5: Figure 1 missing details on PPL/Acc computation.** We clarified that the PPL values are computed on PG19 at 256-token budget, and updated Figure 1 to explicitly state that accuracy values corresponds to the average results on LongBench normalized by full attention, as reported in Table 1 of the revised submission.
>
> - **Q1: Complexity clarification for Adamas vs. full attention.** We clarified that although Adamas has the same asymptotic complexity of $O(L)$ as full attention, the key difference is that Adamas keeps $L$ fixed and constant while the full context length $T$ continues to grow. As a result, $L \ll T$ naturally holds throughout generation. This allows Adamas to reduce the decoding cost from $O(T)$ to $O(L)$, enabling meaningful acceleration.

---

> ### Author Response · Authors · 2025-11-30
> **Summary for the response to weaknesses and questions of Reviewer jitG**
>
> - **W1 & W2: Novelty Misinterpretation of Hadamard Transform and Bucketization.** We clarified that the Hadamard transform **is a classical technique dating back to 1893, and is entirely unrelated to QuaRot.** In our work, it serves a different purpose, dispersing Q/K information to improve the robustness of similarity estimation. We further **emphasized that bucketization is fundamentally distinct from quantization**: it involves no scaling, no dequantization, and produces non-numeric symbolic codes. Finally, we highlighted that Adamas integrates four components into a unified and novel sparse-attention pipeline.
>
> - **W3: Limited comparison to other dynamic/adaptive sparse-attention baselines.** We added SnapKV and L2 as new baselines, explained why MInference is orthogonal (prefill-only) and therefore not directly comparable, and clarified that DUOAttention requires addtional training and has no comparable results. The updated experiments (e.g., Figure 3 and Table 1 in the revised submission)  show that Adamas outperforms all baselines.
>
> - **W4 & Q2: Lack of Theoretical/Emerical Justification for L1 Approximation.** We added a mathematical proof sketch, error bounds for 2-bit bucketization, ordering-preservation analysis, and empirical verification (Figure 3 in the revised submission), demonstrating that L1-based ranking reliably matches dot-product ranking.
>
> - **W5 & Q3: No Analysis of Bucketization Thresholds.** We added threshold selection motivation (Figure 7 in the revised submission), a full sensitivity analysis (Table 6 in the revised submission), comprehensive comparisons showing that thresholds 1 and 10 consistently outperform others (Figure 8 in the revised submission), and threshold selection analysis in Appendix C.
>
> - **Q1: Why Adamas Has Lower PPL Than Full Attention.** We clarified that this effect is consistent with the "lost-in-the-middle" phenomenon reported in prior work: full attention tends to over-distribute weights across long contexts and becomes distracted by large amount of irrelevant tokens. In contrast, Adamas selectively retrieves high-relevance tokens, preventing key evidence from being diluted and thereby achieving better perplexity in long-context settings.
>
> - **Q4: Difference Between Sparse Attention and KV Cache Compression.** We added a clear distinction in Section 5 of the revised submission: sparse attention reduces latency during decoding by selecting fewer tokens for attention computation, while KV-cache compression targets memory reduction but does not necessarily reduce computation, and may even introduce additional overhead. The two approaches address different bottlenecks and are complementary rather than comparable.

---

> ### Author Response · Authors · 2025-11-30
> **Summary for the response to weaknesses and questions of Reviewer W5MJ**
>
> - **W1 & Q2: Missing baselines in efficiency comparison.** We added L2 and SnapKV as additional baselines, provided full LongBench accuracy comparisons (Figure 3 and Table 1 in the revised submission) and A800 end-to-end latency evaluations in Appendix B.
>
> - **W2: Why Adamas matches or surpasses full attention.** We provided theoretical motivation and included a case study showing that full attention suffers from long-context dilution and positional bias ("lost in the middle"), whereas Adamas focuses computation on high-relevance tokens, mitigating these biases and improving perplexity and accuracy.
>
> - **Q1: PPL discrepancy with Quest.** We clarified that Quest originally reported PPL only at a 4096-token budget, under which it performs close to full attention. Adamas consistently outperforms Quest at more challenging token-budgets (256, 512, 1024 and 2048). For completeness, we also provided directed comparison of Adamas and Quest at the 4096-token budget in Appendix D, where Adamas still achieves noticeably lower perplexity than both full attention and Quest.
>
> - **Q3: Why L1 instead of Dot Product.** We explained that L1 distance on 2-bit bucketized codes is computation-efficient (no dequantization required) and provided a full mathematical explanation (ordering preservation via L1/L2 relationships, norm concentration, and error-bounded bucketization), together with empirical evidence confirming strong correlation between L1-based and dot-product rankings.

---

### Meta-Review · Area_Chair_qyNP · 2026-01-08

**Summary:**

The paper proposes  a training-free sparse attention mechanism designed for long-context LLM inference. The paper demonstrates solid engineering work and the authors provided a thorough and professional rebuttal. However, the combination of limited fundamental novelty and the reliance on aggressive heuristics for token selection places this work borderline. The distinction drawn between "bucketization" and standard quantization feels semantic rather than fundamental.

**Reviewer Concerns:**

While the authors present a functional engineering pipeline, the core components are largely applications of existing techniques. The use of the Hadamard transform to reduce quantization error or smooth outliers is well-established in the literature. A significant concern raised by Reviewer jitG, which remains a sticking point, is the use of negative Manhattan distance on 2-bit codes as a proxy for the dot product. While the authors provided a proof sketch and empirical correlation plots during the rebuttal, the theoretical justification relies on strong assumptions regarding norm concentration and distribution. The reliance on a complex pipeline introduces system complexity.

**Reviewer Scores:**

The review process resulted in a split assessment:

Reviewer W5MJ championed the paper (Score: 8) citing the elegant integration and strong results

Reviewer M26R (Score: 6) found the token-level selection promising but had reservations about the evaluation scope

Reviewer jitG (Score: 2) recommended rejection, citing limited novelty and questioning the theoretical grounding of the metrics used

---

### Decision · Program_Chairs · 2026-01-26

Reject